# Beyond the Bellman Recursion: A Pontryagin-Guided Framework for Non-Exponential Discounting

**Hojin Ko** [1]   **Jeonggyu Huh** [1]

## Abstract

Most value-based and actor–critic reinforcement learning methods rely on Bellman-style recursions, yet these recursions collapse under non-exponential discounting common in human preferences and survival processes. We show the breakdown is structural: exponential discounting sits at a fragile intersection of multiplicativity and time homogeneity, and violating either property breaks standard dynamic programming. To overcome this, we propose **Pontryagin-Guided Direct Policy Optimization (PG-DPO)**, a variational framework that abandons recursion and couples the Pontryagin Maximum Principle with Monte Carlo rollouts via an *Adjoint-MC projection* enforcing pointwise Hamiltonian maximization. Across multi-dimensional hyperbolic and survival-discount benchmarks, PG-DPO improves accuracy and stability where equation-driven solvers and critic-based baselines diverge.

## 1. Introduction

Reinforcement learning (RL) and continuous-time stochastic control hinge on a single principle:*Bellman recursion*. It encodes time consistency by requiring that every continuation problem preserves the same objective form. In continuous time, this aligns with *exponential discounting*, which underpins classical HJB equations and many modern deep solvers based on dynamic programming (Bellman, 1957; Sutton & Barto, 2018; Han et al., 2018). However, non-exponential discounting is essential to capture realistic persistent risk in physical systems(Schultheis et al., 2022) and behavioral present bias (Laibson, 1997). Indeed, empirical and theoretical work documents widespread departures from exponential discounting, including hyperbolic

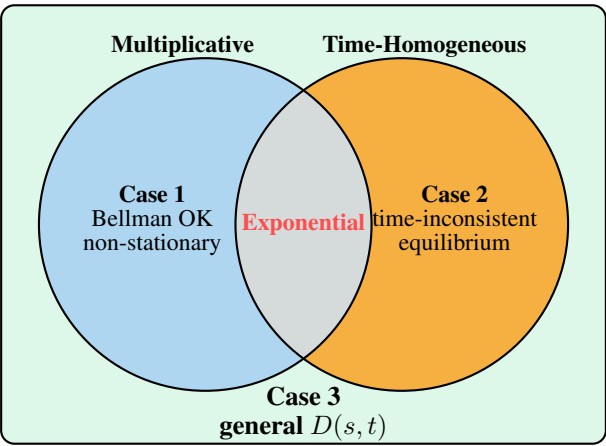

*Figure 1.* **Discount-kernel taxonomy.** Exponential discounting lies at the intersection of multiplicativity (1) and time homogeneity (2). Violating either property invalidates recursion-based methods.

and survival-based patterns (Strotz, 1955; Laibson, 1997; Frederick et al., 2002; Schultheis et al., 2022).

To pinpoint the failure, let $D(s,t)$ denote the discount factor applied at evaluation time $s$ to a payoff realized at time $t \geq s$. Bellman recursion corresponds to a *multiplicativity* of $D$, i.e.,

$$D(s,t) = D(s,u)D(u,t), \qquad \forall s \leq u \leq t. \quad (1)$$

Separately, *time homogeneity* assumes discounting determined only by the delay:

$$D(s,t) = D(s+h, t+h) \qquad \forall s,t,h > 0. \quad (2)$$

Under mild regularity, (1) and (2) together imply the exponential form

$$D(s,t) = e^{-\delta(t-s)} \qquad \text{for some } \delta \geq 0. \quad (3)$$

Thus, "exponential" discounting represents a narrow intersection of these two independent properties, as visualized in Figure 1.

Departing from this exponential corner does not merely generalize the problem; it fundamentally invalidates the core assumptions of the standard pipeline. If $D$ is multiplicative only (Case 1), while recursion technically survives, the failure of stationarity renders standard steady-state solvers obsolete, enforcing the computational burden of time-dependent

[1]Department of Mathematics, Sungkyunkwan University, Suwon, Republic of Korea. Correspondence to: Jeonggyu Huh <jghuh@skku.edu>.

*Proceedings of the 43rd International Conference on Machine Learning*, Seoul, South Korea. PMLR 306, 2026. Copyright 2026 by the author(s).

policies (Schultheis et al., 2022). Conversely, if $D$ is time-homogeneous only (Case 2), the Bellman principle itself collapses, shattering the recursive structure required for dynamic programming. This breakdown forces a shift from optimality to time-consistent equilibrium, a problem typically patched via extended HJB systems (Ekeland & Lazrak, 2006a; Björk & Murgoci, 2014; Yong, 2012) or heuristic value pipelines like UGAE (Kwiatkowski et al., 2023), despite recent efforts to clarify well-posedness (Lei & Pun, 2023; Bayraktar et al., 2025). In Case 3, the pipeline suffers a complete structural failure where both recursion and stationarity are simultaneously lost.

Such failures are intrinsic to recursion-based methods. Existing remedies either try to reinstate recursion through state augmentation or relax optimality to time-consistent equilibrium notions (Ekeland & Lazrak, 2006a; Björk & Murgoci, 2014; Yong, 2012). However, both routes introduce additional time indices and/or nonlocal couplings, which quickly become brittle in high dimensions and undermine the scalability of global equation-driven solvers, including PDE-residual minimization methods such as PINNs (Raissi et al., 2019). Deep BSDE approaches face a closely related obstacle: they rely on global backward representations that are tailored to Bellman-consistent recursions and therefore mismatch non-multiplicative discounting (Han et al., 2018). Indeed, recent progress on deep solvers for genuinely nonlocal backward objects (e.g., BSVIEs) (Andersson et al., 2025) further underscores the need for recursion-free alternatives.

To address this, we adopt a variational approach based on adjoint sensitivity and Pontryagin's maximum principle (PMP) (Chen et al., 2018; Pontryagin et al., 1962), eschewing value-function decomposition entirely (Pontryagin et al., 1962; Yong & Zhou, 1999). While PMP has been revisited for continuous-time RL (Archibald et al., 2023; Eberhard et al., 2025), we propose a unified framework: *Pontryagin-Guided Direct Policy Optimization (PG-DPO)* (Huh et al., 2025).

Our novelty and contributions are: (i) a structural decomposition of recursion failures in non-exponential settings (Figure 1); (ii) a theoretical reformulation of PG-DPO for general discounting; and (iii) empirical gains over *(TD/Bellman-error) actor–critic* baselines and *global equation-driven (surrogate-fitting)* baselines across all non-exponential discounting cases (Cases 1–3).

## 2. Beyond Bellman Optimality: Pontryagin Optimality and PG-DPO

### 2.1. Pontryagin Maximum Principle (PMP)

Non-exponential discounting can invalidate Bellman recursion and thus obstruct a single Markovian value-function characterization. We therefore base our methodology on *Pontryagin optimality* (Pontryagin et al., 1962; Yong & Zhou, 1999). Our viewpoint is *decision-time anchoring*: at each decision time $t_0$, we treat the remaining-horizon problem on $[t_0, T]$ with the explicit kernel $D(t_0, \cdot)$ and enforce a pointwise Pontryagin condition. We record the PMP ingredients that we will enforce algorithmically in PG-DPO.

**General discounted stochastic control.** Fix a finite horizon $[t_0, T]$. Let $(\Omega, \mathcal{F}, (\mathcal{F}_t)_{t \in [t_0, T]}, \mathbb{P})$ support a $d$-dimensional Brownian motion $W_t \in \mathbb{R}^d$. Consider the controlled diffusion

$$dX_t = b(t, X_t, u_t)\, dt + \sigma(t, X_t, u_t)\, dW_t, \quad X_{t_0} = x_0, \tag{4}$$

where $X_t \in \mathbb{R}^d$ and $u_t \in \mathcal{U}$ is an admissible control. Throughout, admissibility means $(u_t)$ is progressively measurable and satisfies the usual integrability conditions that ensure (4) is well-posed.

Let $D : [t_0, T] \times [t_0, T] \to \mathbb{R}_+$ be a discount kernel with $D(s, s) = 1$. For a running reward $\ell$ and terminal reward $g$, define the anchored objective

$$J(t_0, x_0; u) := \mathbb{E}_{t_0, x_0}\Bigg[ \int_{t_0}^{T} D(t_0, t)\, \ell(t, X_t, u_t)\, dt \\ + D(t_0, T)\, g(x_T) \Bigg]. \tag{5}$$

where $\mathbb{E}_{t_0, x_0}[\cdot] := \mathbb{E}[\cdot \mid X_{t_0} = x_0]$.

Under standard smoothness and integrability assumptions, if $u^\star$ denotes the relevant time-consistent solution: when $D$ is multiplicative, we use the term *optimal* $u^* \in \arg\max_{u \in \mathcal{U}} J(t_0, x_0; u)$, otherwise *equilibrium*, characterized by $\liminf_{\epsilon \downarrow 0} \epsilon^{-1}(J(t, s; u^\star) - J(t, s; u_v^\epsilon)) \geq 0$ against any spike variation $u_v^\epsilon$ (Ekeland & Lazrak, 2006b; Björk et al., 2017; Yong, 2012).)

**Anchored Hamiltonian and adjoint BSDE.** For an anchor time $t_0$, define the (anchored) Hamiltonian

$$H(t_0, t, x, u, \lambda, Z) := D(t_0, t)\, \ell(t, x, u) + \langle \lambda, b(t, x, u) \rangle \\ + \mathrm{Tr}\big(Z^\top \sigma(t, x, u)\big). \tag{6}$$

where $\lambda_t \in \mathbb{R}^d$ is the adjoint process and $Z_t \in \mathbb{R}^{d \times d}$ is the diffusion coefficient in the backward equation.

Let $X^\star$ be the induced state trajectory from $u^\star$. Then, there exist adapted processes $(\lambda^\star, Z^\star)$ satisfying the adjoint equation:

$$-d\lambda_t^\star = \partial_x H\big(t_0, t, X_t^\star, u_t^\star, \lambda_t^\star, Z_t^\star\big)\, dt - Z_t^\star\, dW_t, \\ \lambda_T^\star = D(t_0, T)\, \nabla g\big(X_T^\star\big). \tag{7}$$

**Maximum condition.** PMP enforces a pointwise maximization of the Hamiltonian (Pontryagin et al., 1962; Yong & Zhou, 1999):

$$u_t^\star \in \arg\max_{u \in \mathcal{U}} H\big(t_0, t, X_t^\star, u, \lambda_t^\star, Z_t^\star\big). \tag{8}$$

This condition holds for a.e. $t \in [t_0, T]$, $\mathbb{P}$-a.s.

**Implications of PMP.** The Maximum Condition (8) implies that the optimal/equilibrium control $u^\star$ is directly determined by the adjoint $\lambda^\star$, meaning that estimating $\lambda^\star$ is sufficient to synthesize $u^\star$.

Existing deep solvers typically do auxiliary function approximation to estimate $\lambda^\star$, based on Bellman recursion (HJB, BSDE). Even under time-inconsistency, this paradigm is rigidly maintained via Extended HJB systems or BSVIEs, despite the loss of standard recursion.

However, as is well known, global function approximation is often sensitive to heuristics and environmental stochasticity. Furthermore, the inherent recursive structure suffers from error propagation, where initial approximation errors inevitably accumulate step-by-step.

We propose a fundamentally different algorithmic paradigm: we interpret Backpropagation Through Time (BPTT) as a *stochastic adjoint estimator*[1]. In our framework, (8) plays the role of a Bellman-free *time-consistent* condition and will be enforced by the Adjoint-MC projection step in Section 2.2.

### 2.2. Pontryagin-Guided Direct Policy Optimization

PG-DPO is a two-stage, Bellman-free procedure that couples Monte Carlo rollouts with a Pontryagin projection. Beyond prior PG-DPO(Huh et al., 2025), tailored to time-consistent objectives, we enforce the decision-time–anchored (diagonal) Pontryagin condition to obtain a time-consistent equilibrium under non-exponential discounting. Stage 1 warm-starts a differentiable policy by direct rollout optimization; Stage 2 estimates $\lambda$ via BPTT and performs action-space Hamiltonian maximization.

**Policy class.** We parameterize the control as a neural network $u_\theta(t, x)$ with explicit time input. This network does not serve as a final output of policy, but only intermediate proxy to guide state exploration.

**Discretization and Monte Carlo objective.** For computing integrals by finite sums, we discretize $[t_0, T]$ over short time steps with $t_k = t_0 + k\Delta t$, $\Delta t = (T - t_0)/N$. Using Euler–Maruyama,

$$
\begin{aligned}
X_{k+1} = X_k &+ b\big(t_k, x_k, u_\theta(t_k, x_k)\big)\, \Delta t \\
&+ \sigma\big(t_k, x_k, u_\theta(t_k, X_k)\big)\, \Delta W_k,
\end{aligned} \tag{9}
$$

with $\Delta W_k \sim \mathcal{N}(0, \Delta t)$. For each simulated path, we ap-

---

[1]We assume access to a stochastic simulator (physics-based or statistically estimated), or a differentiable learned world model, that supports pathwise differentiation and thus enables BPTT-based adjoint estimation.

---

**Algorithm 1** PG-DPO

1: **Input:** policy $u_\theta$; anchor dist. $\nu$; rollout steps $N$; batch $M$; iters $K_0$; stepsizes $\{\alpha_j\}$.
2:     projection params $(M_{\mathrm{MC}}, N')$; query set $\mathcal{Q}$ (each $(t, x)$).
3: **Stage 1 (rollout warm-start):** initialize $\theta$
4: **for** $j = 0, \ldots, K_0 - 1$ **do**
5:     Sample $\{(t_0^{(i)}, x_0^{(i)})\}_{i=1}^M \sim \nu$ and simulate $M$ rollouts via (9)
6:     $\widehat{g} \leftarrow \frac{1}{M} \sum_{i=1}^M \nabla_\theta \widehat{J}(t_0^{(i)}, x_0^{(i)}; \theta)$       (BPTT)
7:     $\theta \leftarrow \theta + \alpha_j \widehat{g}$
8: **end for**
9: $\theta^\star \leftarrow \theta$
10: **Stage 2 (Adjoint-MC projection):**
11: **for** each $(t, x) \in \mathcal{Q}$ **do**
12:     Simulate $M_{\mathrm{MC}}$ rollouts from $(t, x)$ under $u_{\theta^\star}$ (horizon $N'$; anchor at $t$)
13:     $\widehat{\lambda}(t, x) \leftarrow \frac{1}{M_{\mathrm{MC}}} \sum_{m=1}^{M_{\mathrm{MC}}} \frac{\partial \widehat{J}^{(m)}(t, x; \theta^\star)}{\partial x}$    (BPTT)
14:     Compute $\hat{u}(t, x)$ via (12) (few Newton/barrier steps; warm-start at $u_{\theta^\star}(t, x)$)
15: **end for**
16: **Return:** $\theta^\star$ and $\hat{u}$ on $\mathcal{Q}$

---

proximate (5) by

$$
\begin{aligned}
\widehat{J}(t_0, x_0; \theta) := \sum_{k=0}^{N-1} & D(t_0, t_k)\, \ell\big(t_k, X_k, u_\theta(t_k, X_k)\big)\, \Delta t \\
&+ D(t_0, T)\, g(X_N).
\end{aligned} \tag{10}
$$

**Stage 1: rollout warm-start.** We warm-start $\theta$ by maximizing $\widehat{J}(t_0, x_0; \theta)$ via BPTT on the rollout graph, optionally randomizing anchors $(t_0, x_0) \sim \nu$.

**Stage 2: Adjoint-MC projection (control synthesis).** Given a query $(t, x)$ and the frozen warm-start policy $u_{\theta^\star}$, we discretize $[t, T]$ by $t_k^{(t)} = t + k\Delta t'$, $\Delta t' = (T - t)/N'$, simulate $M_{\mathrm{MC}}$ rollouts from $(t, x)$, and evaluate an objective *anchored* at $t$:

$$
\begin{aligned}
\widehat{J}^{(j)}(t, x; \theta^\star) = \sum_{k=0}^{N'-1} & D(t, t_k^{(t)})\ell\big(t_k^{(t)}, X_k^{(j)}, u_{\theta^\star}(t_k^{(t)}, X_k^{(j)})\big)\Delta t' \\
&+ D(t, T)\, g\big(X_{N'}^{(j)}\big).
\end{aligned}
$$

For each rollout $j$, we compute a pathwise costate by BPTT, $\lambda^{(j)}(t, x) := \partial \widehat{J}^{(j)}(t, x; \theta^\star)/\partial x$, and average

$$
\widehat{\lambda}(t, x) := \frac{1}{M_{\mathrm{MC}}} \sum_{j=1}^{M_{\mathrm{MC}}} \lambda^{(j)}(t, x). \tag{11}
$$

Anchoring at $t$ yields the diagonal Pontryagin condition (anchor = decision time) used for equilibrium control when multiplicativity fails; see Section 2.3.

We then finally compute the pointwise Hamiltonian maximizer (8)

$$\hat{u}(t,x) \in \arg\max_{u \in \mathcal{U}(x)} H\big(t, t, x, u, \widehat{\lambda}(t,x), \widehat{Z}(t,x)\big), \quad (12)$$

In many problems the maximizer depends only on $\widehat{\lambda}$ (so $\widehat{Z}$ is not needed); when $Z$ is required, it can be estimated by a standard one-step $L^2$ projection/regression on $\Delta W$.

**Computing the projection (Newton / log-barrier).** Equation (12) does not require a closed form in constraint problem: we solve the action-space maximization by a few warm-started Newton (or quasi-Newton) iterations. With inequality constraints $\mathcal{U}(x) = \{u : g_i(u,x) \leq 0\}$, we use the interior-point objective

$$\max_u\ H\big(t,t,x,u,\widehat{\lambda}(t,x),\widehat{Z}(t,x)\big) + \mu \sum_i \log\big(-g_i(u,x)\big),$$
$$(13)$$

together with backtracking line search to maintain feasibility. Optionally, this projection step can be amortized via offline distillation of $\hat{u}(t,x)$ into a student policy $\pi_\phi(t,x)$ for faster deployment.

### 2.3. BPTT as a stochastic adjoint estimator

A key reason PG-DPO remains effective under non-exponential discounting is that it does *not* learn a value function (critic) or rely on Bellman recursion. Instead, it computes *value-gradient information on-the-fly* from differentiable Monte Carlo rollouts, and then turns this gradient into an action by enforcing a Pontryagin (Hamiltonian) condition.

**(1) BPTT gives *marginal value* with respect to the state.** Fix an anchor time $t_0$ and consider the anchored rollout return $\widehat{J}(t_0, s_0; \theta)$ in (10). Reverse-mode differentiation through the rollout graph produces

$$\lambda_k^{\mathrm{pw}} := \frac{\partial \widehat{J}(t_0, x_0; \theta)}{\partial X_k},$$

which measures how much the anchored objective would change under an infinitesimal perturbation of the state at time $t_k$. In continuous-time PMP language, this is exactly the *costate* interpretation: $\lambda$ is the marginal value of the state (Pontryagin et al., 1962; Yong & Zhou, 1999).

**(2) Why averaging is the point (variance reduction & adaptedization).** A single pathwise gradient $\lambda^{\mathrm{pw}}$ is noisy because it depends on one Brownian realization. Stage 2 therefore computes *stabilized* costate estimates by Monte Carlo averaging of BPTT state-gradients,

$$\widehat{\lambda}(t,x) := \frac{1}{M_{\mathrm{MC}}} \sum_{j=1}^{M_{\mathrm{MC}}} \lambda^{(j)}(t,x), \quad (14)$$

as already defined in (11). Intuitively, $\widehat{\lambda}(t,x)$ plays the role of a *value-gradient critic* evaluated at $(t,x)$, but obtained by autodiff through the simulator rather than by training a separate network.

**(3) How averaged BPTT sensitivities ($\widehat{\lambda}$) become anchored/diagonal adjoint proxies.** Because BPTT differentiates through the feedback dependence $u_k = u_\theta(t_k, X_k)$, the exact state-sensitivity recursion contains an extra closed-loop term:

$$\lambda_k^{\mathrm{pw}} = \partial_{X_k} r_k + (\partial_{X_k} F_k)^\top \lambda_{k+1}^{\mathrm{pw}} + (\partial_{X_k} u_k)^\top G_k,$$

where $G_k$ is a discrete stationarity residual (a discrete analogue of $\partial_u H$; Appendix A). The key point is that this recursion should be matched not to an unknown *optimal* costate, but to the *exact anchored adjoint of the warm-up policy itself*. After taking the predictable projection and separating the policy-sensitivity term, the adapted BPTT sensitivity satisfies the discretized anchored adjoint drift identity up to (i) an explicit predictable Hamiltonian-residual correction and (ii) the standard one-step Euler remainder. Specializing the anchor to the decision time gives the diagonal costate proxy used by Stage 2. The following theorem makes this correspondence precise.

**Theorem 2.1** (Anchored costate–BPTT correspondence). *Fix an anchor $t_0$ and the Stage 1 warm-up policy $u_{\theta^\star}$. Let $X_k$ be the Euler rollout and set $u_k := u_{\theta^\star}(t_k, X_k)$. Define*

$$\lambda_k^{\mathrm{pw}} := \frac{\partial \widehat{J}(t_0, x_0; \theta^\star)}{\partial X_k}, \quad \lambda_k := \mathbb{E}_k[\lambda_k^{\mathrm{pw}}],$$

$$\lambda_{k+1|k} := \mathbb{E}_k[\lambda_{k+1}^{\mathrm{pw}}], \qquad Z_k := \frac{1}{\Delta t} \mathbb{E}_k[\lambda_{k+1}^{\mathrm{pw}} \Delta W_k^\top]. \quad (15)$$

*For $p$ and $Z$ of compatible dimensions, write*

$$H_k^{t_0}(x,u;p,Z) := D(t_0, t_k)\ell(t_k, x, u) + p^\top b(t_k, x, u) + \langle Z, \sigma(t_k, x, u)\rangle_F. \quad (16)$$

*Along the Stage 1 rollout, define*

$$D_k^x := \partial_x H_k^{t_0}(X_k, u_k; \lambda_{k+1|k}, Z_k),$$
$$r_k^{\mathrm{FOC}} := \partial_u H_k^{t_0}(X_k, u_k; \lambda_{k+1|k}, Z_k), \quad (17)$$
$$\mathcal{C}_k^{\mathrm{FOC}} := (r_k^{\mathrm{FOC}})^\top \partial_x u_{\theta^\star}(t_k, X_k).$$

*Under the standing assumptions in Appendix A, there exists an $\mathcal{F}_{t_k}$-measurable remainder $\rho_k^{\mathrm{disc}}$ satisfying $\|\rho_k^{\mathrm{disc}}\|_{L^2} = o(\Delta t)$ such that*

$$\lambda_k = \lambda_{k+1|k} + \big(D_k^x + \mathcal{C}_k^{\mathrm{FOC}}\big)\Delta t + \rho_k^{\mathrm{disc}}. \quad (18)$$

*Thus, the gap between adapted BPTT sensitivities and the Euler discretization of the anchored adjoint is given by the predictable Hamiltonian residual $\mathcal{C}_k^{\mathrm{FOC}}\Delta t$, up to the usual one-step discretization error.*

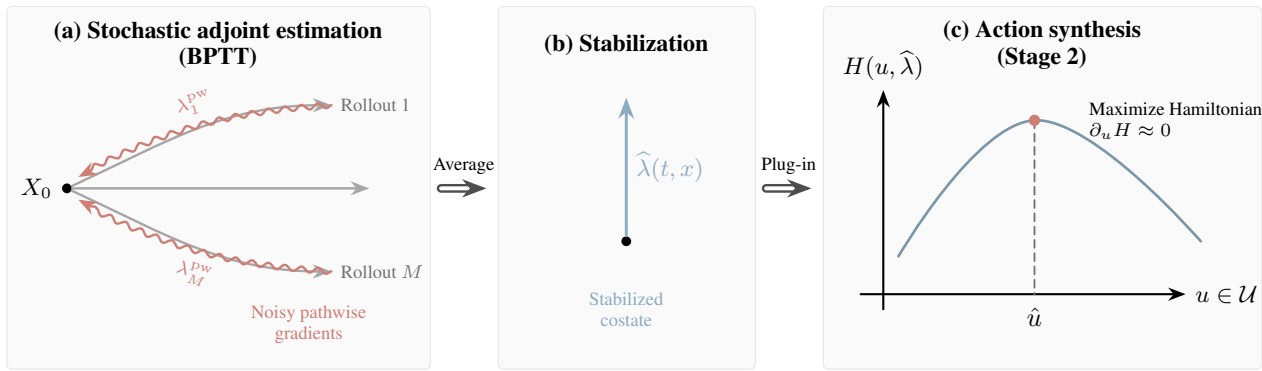

*Figure 2.* **Mechanism of Adjoint-MC Projection.** (a) BPTT computes noisy pathwise state-gradients ($\lambda^{\mathrm{pw}}$) from anchored rollouts. (b) Monte Carlo averaging stabilizes these gradients into a robust costate estimate $\widehat{\lambda}(t, x)$. (c) This estimate defines the local Hamiltonian $H(\cdot, \widehat{\lambda})$, which is maximized in action space to synthesize $u^{\mathrm{proj}}$, enforcing the Pontryagin condition directly.

*Moreover, if*

$$\|\partial_x u_{\theta^\star}(t_k, X_k)\|_{L^\infty} \le C_u, \qquad \|r_k^{\mathrm{FOC}}\|_{L^2} \le \varepsilon_k^{\mathrm{FOC}}, \tag{19}$$

*then*

$$\|\mathcal{C}_k^{\mathrm{FOC}}\|_{L^2} \le C_u\, \varepsilon_k^{\mathrm{FOC}},$$
$$\|\lambda_k - \lambda_{k+1|k} - D_k^x \Delta t\|_{L^2} \le C_u\, \varepsilon_k^{\mathrm{FOC}} \Delta t + o(\Delta t). \tag{20}$$

*In particular, exact predictable stationarity makes the adapted BPTT sensitivities coincide with the discretized anchored adjoint up to the standard Euler error. The martingale one-step representation used in the proof is given in Appendix A.4. Specializing the anchor to the decision time, $t_0 = t_k$, yields the diagonal costate proxy used by Stage 2.*

**(4) Stage 2: action synthesis by (approximately) killing the Hamiltonian residual.** Stage 2 is an action-synthesis step: it constructs a *new* control $u^{\mathrm{proj}}(t, x)$ by maximizing the Hamiltonian anchored at the decision time $t$,

$$u^{\mathrm{proj}}(t, x) \in \arg\max_{u \in \mathcal{U}(x)} H\big(t, t, x, u, \widehat{\lambda}(t, x), \widehat{Z}(t, x)\big).$$

In this sense, Stage 2 enforces a near-zero Hamiltonian stationarity residual *with respect to the plug-in costate estimate* at the query point,

$$\partial_u H\big(t, t, x, u^{\mathrm{proj}}(t, x), \widehat{\lambda}(t, x), \widehat{Z}(t, x)\big) \approx 0.$$

This can be viewed as a Pontryagin-style *policy-improvement* step: given a costate estimate, we synthesize an action that is stationary for the corresponding anchored Hamiltonian.

A rigorous local control-proximity guarantee for this projection step is stated in Theorem 2.2 below and proved in Appendix B.

### 2.4. Technical Advantage

**Rigorous stability of Stage 2.** The diagonal Pontryagin projection is not merely a heuristic post-processing step. The next result summarizes the rigorous Stage 2 guarantee proved in Appendix B.

**Theorem 2.2** (Rigorous local control-proximity guarantee for Stage 2)**.** *Let $u^{\mathrm{proj}}$ denote the Stage 2 projected control and let $u^\star$ denote the exact diagonal Pontryagin control, i.e., the classical optimal control when $D$ is multiplicative and the diagonal equilibrium control otherwise. Write $\|\cdot\|_{\ell,2}$, $r^{\mathrm{warm}}$, $\delta_\ell^{\mathrm{diag}}$, and $\eta_\ell^{\mathrm{num}}$ for the slab norms and residual/error terms defined in Appendix B. Under the local projection regularity, short-slab propagation, residual-to-solution transfer, and finite patching conditions stated there, there exist constants $\tau_0, C_{\mathrm{diag}} > 0$ such that for any slab partition with length $\tau \le \tau_0$,*

$$\sum_{\ell=0}^{L-1} \|u^{\mathrm{proj}} - u^\star\|_{\ell,2} \le C_{\mathrm{diag}} \sum_{\ell=0}^{L-1} \Big( \|r^{\mathrm{warm}}\|_{\ell,2} + \delta_\ell^{\mathrm{diag}} + \eta_\ell^{\mathrm{num}} \Big), \tag{21}$$

*where $\delta_\ell^{\mathrm{diag}}$ is the diagonal adjoint-estimation error and $\eta_\ell^{\mathrm{num}}$ is the numerical projection error on slab $I_\ell$. Hence, if the warm-up residual is small, the diagonal BPTT adjoint estimate is accurate, and the pointwise Hamiltonian solve is numerically tight, then Stage 2 remains close to the exact diagonal control. Appendix B gives the full quantitative statement, including the multiplicative value-gap consequence and the non-multiplicative equilibrium-defect consequence.*

### Bypassing the Global Approximation Bottleneck.

Existing deep RL/solvers generally struggle with a trade-off between computational cost and control precision. Purely end-to-end actor–critic training is often brittle because it uses a global averaged objective to enforce a *pointwise*

stationarity condition ($\partial_u H \approx 0$), so satisfying this local condition uniformly over the state space can require excessive iterations. Recent *model-based* trends partially mitigate sample inefficiency by learning a stochastic "world model" and planning through it; however, the difficulty of turning global training signals into uniformly accurate pointwise optimality conditions remains.

*Global equation-driven* approaches (e.g., PINNs, Deep BS-DEs) face a different bottleneck: they rely on global function approximation. A small error in the surrogate does not guarantee accurate recovery of its gradients, so expensive training can still yield unreliable controls.

PG-DPO fundamentally breaks this dependency. Even if the Stage 1 policy provides only a coarse approximation (weak optimality), the Stage 2 projection effectively minimizes the Hamiltonian residual via direct optimization (Appendix E). Theorem 2.2 and Appendix B make the resulting control-proximity guarantee explicit, so PG-DPO achieves high-confidence control synthesis with low training cost without requiring perfect global approximation.

**Computational Efficiency.** For a *single* query $(t, x)$, the dominant cost of Stage 2 is linear in the total number of simulated steps,

$$T(t, x) = \mathcal{O}\Big( M_{\mathrm{MC}} \, N' \, C_{\mathrm{step}} \;+\; I_{\mathrm{proj}} \, C_{\mathrm{proj}} \Big), \quad (22)$$

where $C_{\mathrm{step}}$ is the per-step cost of evaluating the frozen warm-start policy $u_{\theta^\star}$ and propagating the Euler–Maruyama dynamics. $C_{\mathrm{proj}}$ is the cost per optimization iteration (e.g., evaluating the Hamiltonian gradient and Hessian), and $I_{\mathrm{proj}}$ is the number of Newton/quasi-Newton (or barrier) iterations used to solve (12). In typical applications $I_{\mathrm{proj}}$ is a small constant (e.g., a handful of warm-started steps), so the overall runtime is dominated by costate estimation (BPTT). Our empirical experiments in Appendix G validate this linear scaling, demonstrating that our method achieves sub-second inference latency ($< 0.02$s) for our base configuration even on commodity CPUs. This efficiency implies that PG-DPO is not merely an offline solver but is computationally viable for online real-time control synthesis, where rapid re-planning is critical.

If even lower latency is desired, the per-query Stage 2 projection can be amortized offline by distilling the synthesized actions $\hat{u}(t, x)$ over a representative query set into a lightweight student policy, reducing deployment to a single forward pass.

## 3. Numerical Results

Our experiments cover the entire spectrum of discount-kernel regimes depicted in Figure 1. This confirms that **PG-DPO** generalizes effectively across diverse non-exponential

discounting landscapes, verifying its comprehensive applicability. Throughout, we include multi-dimensional setting ($d = 5$) and visualize only the first control coordinate ($u_1$ or $\pi_1$) for clarity. For scalability, high-dimensional settings ($10 \leq d \leq 100$) are reported in the Appendix F.

Baselines are chosen to span increasing levels of model information: PPO (Schulman et al., 2017) as a model-free actor–critic baseline, DPO (Stage I) as a simulator-level policy-optimization baseline, and PINN (Raissi et al., 2019) and Deep BSDE/BSVIE (Han et al., 2018) as global equation-driven baselines; detailed definitions and interpretation guidelines are deferred to Appendix D.[2]

We demonstrate the high accuracy and robustness of **PG-DPO** by reporting the $L_1$ error and standard deviation across multiple random seeds. To ensure that the observed performance gap is purely algorithmic, we allocated a larger computational budget and model complexity exclusively to the approximation step of the baselines. [3]

### 3.1. Case 1 Benchmark: Survival-Discounted Target Control

**Discounting and task.** Following Schultheis et al. (2022) (see also Aalen et al., 2008), we model discounting via survival under a Gamma prior on the hazard rate:

$$S(t) = \left( \frac{\beta_0}{\beta_0 + t} \right)^{\alpha_0}, \quad D(s, t) = \frac{S(t)}{S(s)} = \left( \frac{\beta_0 + s}{\beta_0 + t} \right)^{\alpha_0},$$

for $0 < s < t$. This preserves multiplicativity but destroys stationarity. In Case 1, the agent drives a controlled diffusion toward a target with a quadratic control-energy penalty. The survival kernel admits a natural *random termination* interpretation: when survival drops rapidly early on (Figure 3(a), small $\beta_0$), the objective becomes urgent and explicitly time-inhomogeneous. Overall, Case 1 should be viewed as an abstraction aligned with standard continuous-control objectives (stabilization/goal reaching/tracking under termination), while isolating the effect of non-stationary discounting.

**Why PG-DPO succeeds.** Figure 3(b) confirms the superiority of our approach. Panel (a) shows that PG-DPO tracks the ground-truth policy almost perfectly, and the very narrow Min–Max band indicates strong robustness across random seeds. In contrast, competing solvers (b–d) exhibit

---

[2]Grid-based DP is omitted because its computational cost scales exponentially with dimension. *Ground Truth* denotes reference policies from the standard HJB in the multiplicative regime (Case 1) or from the extended/equilibrium HJB in the time-inconsistent regimes (Cases 2–3).

[3]For additional experimental details, see Appendix H. For full details and reproducibility, see https://github.com/Non-exponential/ Beyond-the-Bellman-Recursion

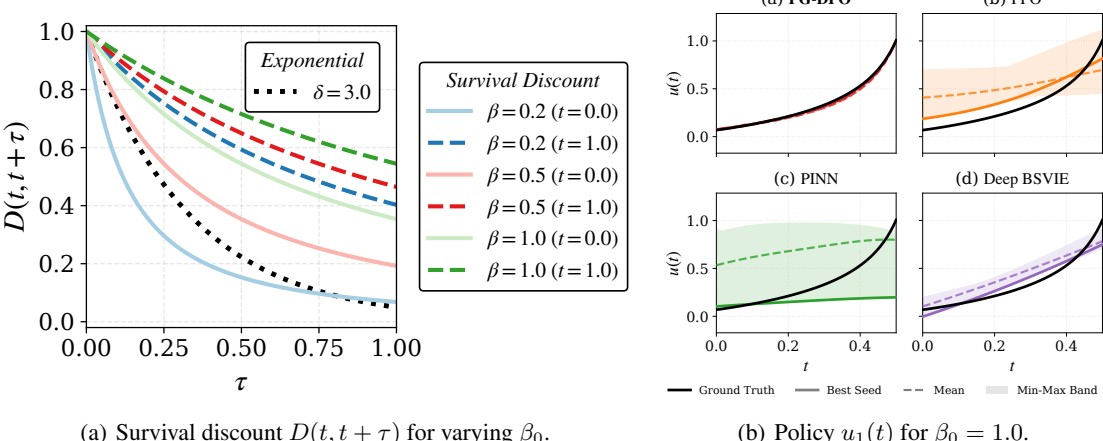

(a) Survival discount $D(t, t+\tau)$ for varying $\beta_0$.

(b) Policy $u_1(t)$ for $\beta_0 = 1.0$.

*Figure 3.* **Case 1 (survival discounting).** (a) The survival-based kernel is multiplicative but time-inhomogeneous. (b) Learned controls compared to the analytic policy along a representative trajectory.

*Table 1.* **Quantitative comparison of control policy errors.** Values are reported as Mean $\pm$ Std over 10 seeds.

| | **Global $L_1$ Error** $(\iint |u^\star - \hat{u}| dx dt)$ | | |
|---|---|---|---|
| **Method** | $\beta_0 = 0.2$ | $\beta_0 = 0.5$ | $\beta_0 = 1.0$ |
| PPO | $2.51\mathrm{e}{-1} \pm 4.48\mathrm{e}{-2}$ | $2.18\mathrm{e}{-1} \pm 5.06\mathrm{e}{-2}$ | $2.23\mathrm{e}{-1} \pm 4.90\mathrm{e}{-2}$ |
| PINN | $1.11\mathrm{e}0 \pm 2.07\mathrm{e}{-1}$ | $1.16\mathrm{e}0 \pm 1.20\mathrm{e}{-1}$ | $1.65\mathrm{e}0 \pm 2.53\mathrm{e}{-1}$ |
| Deep BSDE | $1.95\mathrm{e}{-1} \pm 4.59\mathrm{e}{-2}$ | $3.71\mathrm{e}{-1} \pm 2.90\mathrm{e}{-2}$ | $4.17\mathrm{e}{-1} \pm 1.52\mathrm{e}{-1}$ |
| DPO | $3.82\mathrm{e}{-2} \pm 1.14\mathrm{e}{-2}$ | $6.15\mathrm{e}{-2} \pm 1.87\mathrm{e}{-2}$ | $4.42\mathrm{e}{-2} \pm 1.36\mathrm{e}{-2}$ |
| **PGDPO (Ours)** | $\mathbf{1.45}\mathrm{e}{-}\mathbf{2} \pm \mathbf{5.14}\mathrm{e}{-}\mathbf{3}$ | $\mathbf{2.97}\mathrm{e}{-}\mathbf{2} \pm \mathbf{1.29}\mathrm{e}{-}\mathbf{2}$ | $\mathbf{1.80}\mathrm{e}{-}\mathbf{2} \pm \mathbf{8.10}\mathrm{e}{-}\mathbf{3}$ |

either large variance (PPO, PINN) or systematic bias (Deep BSDE), failing to match the correct curvature. Table 1 reports the quantitative comparison along $\beta_0$. PG-DPO attains the lowest $L_1$ error with the smallest standard deviation. This gap arises because Stage 2 enforces the Pontryagin condition locally in time: decision-time–anchored rollouts yield a costate that captures steep early discounting, and Hamiltonian maximization maps this local marginal value to an action without requiring a globally consistent value-function fit.

### 3.2. Case 2 Benchmark: Merton Problem with Hyperbolic Discounting

**Discounting and task.** We use the classical hyperbolic kernel (Strotz, 1955; Laibson, 1997; Frederick et al., 2002)

$$D(s, t) = \frac{1}{1 + \kappa(t - s)} \qquad (\tau := t - s).$$

This kernel is time-homogeneous but non-multiplicative, hence time-inconsistent. Accordingly, performance is evaluated against the time-consistent (equilibrium) solution of the extended HJB (Ekeland & Lazrak, 2006a; Björk & Murgoci,

2014; Yong, 2012).

**Why PG-DPO succeeds.** In this case, restoring timehomogeneity simplifies the control problem to a stationary regime, generally reducing variance across all baselines compared to the non-stationary setting (Case 1). Despite this shared benefit, a clear performance hierarchy remains evident. As shown in Figure 4, *global equation-driven* baselines (PINN, Deep BSVIE) outperform the *critic-based (TD/Bellman-error) actor–critic* baseline (PPO), while PG-DPO achieves near-perfect alignment with the ground truth for both consumption and investment policies, exhibiting overwhelming accuracy and significantly tightest confidence bands. This superiority is quantitatively confirmed in Table 2, where PG-DPO reduces $L_1$ and $L_\infty$ errors by several orders of magnitude. This precise alignment stems from Stage 2, which leverages the stationary structure to stabilize local costates via anchored rollouts, allowing Hamiltonian maximization to synthesize the exact equilibrium action.

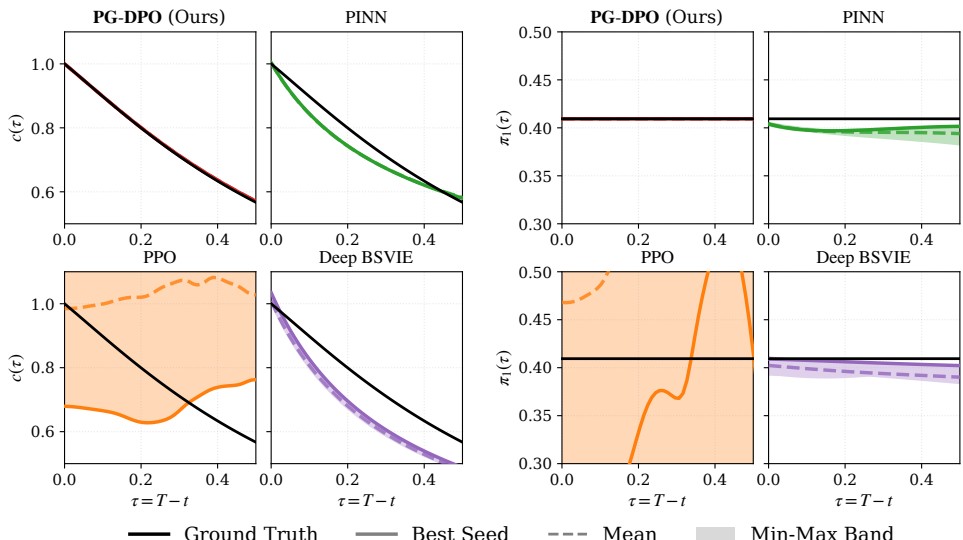

*Figure 4.* **Case 2 equilibrium policies.** Semi-analytic (extended-HJB) equilibrium vs. learned controls. (Left 2x2: Consumption Policy / Right 2x2: Investment Policy)

*Table 2.* **Quantitative comparison of errors**. The table reports Global $L_1, L_\infty$ error as Mean $\pm$ Std over 10 seeds.

| Method | Consumption ($c$) | | Investment ($\pi$) | |
|---|---|---|---|---|
| | **MAE ($L_1$)** | **Max ($L_\infty$)** | **MAE ($L_1$)** | **Max ($L_\infty$)** |
| PPO | 4.66e-01 $\pm$ 1.35e-01 | 1.03e+00 $\pm$ 2.79e-01 | 5.39e-01 $\pm$ 6.34e-02 | 2.22e+00 $\pm$ 4.03e-01 |
| PINN | 6.67e-02 $\pm$ 3.01e-03 | 2.17e-01 $\pm$ 1.83e-02 | 4.73e-02 $\pm$ 5.95e-03 | 3.05e-01 $\pm$ 2.57e-02 |
| BSVIE | 6.41e-02 $\pm$ 4.86e-03 | 1.46e-01 $\pm$ 1.13e-02 | 4.08e-02 $\pm$ 5.01e-03 | 2.44e-01 $\pm$ 1.25e-02 |
| DPO | 2.32e-01 $\pm$ 1.18e-01 | 7.85e-01 $\pm$ 2.45e-01 | 3.12e-01 $\pm$ 5.92e-02 | 1.11e+00 $\pm$ 3.78e-01 |
| **PG-DPO** | **3.47**e-03 $\pm$ 2.41e-10 | **6.11**e-03 $\pm$ 1.19e-08 | **6.36**e-08 $\pm$ 3.37e-10 | **1.92**e-06 $\pm$ 1.46e-07 |

### 3.3. Case 3 Benchmark: Stochastic Resource with Time-Varying Impatience

**Discounting and task.** We generalize Case 2 by allowing the hyperbolic parameter to vary with the decision time (Björk & Murgoci, 2014; Yong, 2012):

$$D(s,t) = \frac{1}{1 + k(s)(t-s)}.$$

This kernel is non-multiplicative and explicitly non-stationary, inducing time inconsistency. Case 3 considers a stochastic resource/consumption problem where impatience fluctuates over time.

**Why PG-DPO succeeds.** Figure 5 shows that equilibrium policy becomes *non-monotone* and co-moves with $k(t)$. PG-DPO tracks these regime changes because Stage 2 is decision-time anchored: when $k(t)$ changes, the anchored kernel $D(t, \cdot)$ and the corresponding Hamiltonian change immediately, and the action is synthesized by local Hamiltonian maximization rather than by relying on a globally trained value function. Across the whole $k(t)$, Table 3

shows uniformly smallest errors, supporting stable recovery of the time-consistent equilibrium mechanism under explicit non-stationarity.

**Takeaway.** Taken together with Cases 2–3, the results support the central message: when multiplicativity fails, PG-DPO benefits from Stage 2 by enforcing the decision-time Pontryagin condition through local action synthesis.

## 4. Conclusion

We showed that the ubiquity of exponential discounting in RL is structural: it arises from the intersection of multiplicativity and time homogeneity (Figure 1; Equations (1) to (3)). Violating either property breaks the recursive machinery behind dynamic programming and single-time BSDE formulations, helping explain the empirical instability of standard methods in non-exponential settings.

To bridge this gap, we introduced Pontryagin-Guided Direct Policy Optimization (PG-DPO; Section 2.2). PG-DPO is a model-based, simulator-access method tailored to settings

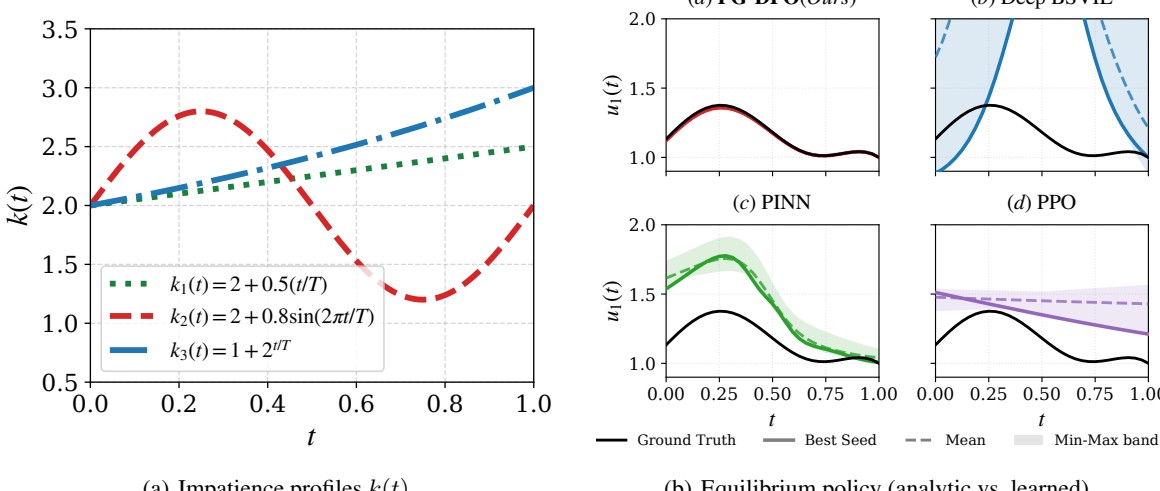

(a) Impatience profiles $k(t)$.

(b) Equilibrium policy (analytic vs. learned).

*Figure 5.* **Case 3 (time-varying hyperbolic discounting).** (a) Time-varying impatience profiles $k(t)$ we used. (b) Equilibrium consumption under non-stationary discounting in case of $k_2(t)$.

*Table 3.* **Quantitative comparison of errors (Case 3).** The table reports Global $L_1$ as Mean $\pm$Std under three different time-varying impatience profiles $k(t)$.

| Method | Linear | Sinusoidal | Exponential |
|---|---|---|---|
| Deep BSVIE | $1.29\text{e}0 \pm 3.85\text{e-}01$ | $1.02\text{e}0 \pm 2.30\text{e-}01$ | $1.36\text{e}0 \pm 4.11\text{e-}01$ |
| PINN | $2.64\text{e-}01 \pm 3.60\text{e-}02$ | $2.67\text{e-}01 \pm 3.66\text{e-}02$ | $2.93\text{e-}01 \pm 5.97\text{e-}02$ |
| PPO | $2.68\text{e-}01 \pm 5.84\text{e-}02$ | $2.89\text{e-}01 \pm 5.14\text{e-}02$ | $2.42\text{e-}01 \pm 5.62\text{e-}02$ |
| DPO | $1.15\text{e-}01 \pm 4.12\text{e-}02$ | $1.31\text{e-}01 \pm 3.95\text{e-}02$ | $1.08\text{e-}01 \pm 4.30\text{e-}02$ |
| **PG-DPO** | $\mathbf{6.35}\text{e-}\mathbf{03} \pm \mathbf{2.51}\text{e-}\mathbf{05}$ | $\mathbf{7.39}\text{e-}\mathbf{03} \pm \mathbf{3.88}\text{e-}\mathbf{05}$ | $\mathbf{6.70}\text{e-}\mathbf{03} \pm \mathbf{3.19}\text{e-}\mathbf{05}$ |

with a stochastic simulator (physics-based or statistically estimated). By interpreting BPTT as a stochastic adjoint estimator, PG-DPO recovers costate information without a Markovian value function and replaces broken Bellman recursion with a variational principle.

Our approach advances the optimization landscape in two views. First, it achieves structural flexibility by liberating the optimization process from the Bellman recursion with mathematical consistency. Second, it offers technical robustness by moving beyond the standard frame of function approximation. This shift significantly reduces the reliance on heuristic tuning and stochastic environment.

**Limitations and future work.** The method requires a differentiable simulator or learned dynamics model, and projection quality depends on accurate BPTT marginal values. Looking forward, promising directions include variance reduction, amortized projection, learned dynamics, empirical calibration, relaxing regularity to accommodate data-driven or non-smooth discount kernels and extending the projection machinery to richer constraints and frictions.

## Impact Statement

This paper introduces PG-DPO, a method for optimizing policies under non-exponential discounting. Our work is primarily methodological, aiming to bridge the gap between theoretical control and practical applications. While effective control algorithms can have broad societal impacts, we believe this specific work does not introduce new ethical concerns beyond those already present in the fields of reinforcement learning and optimal control.

## Acknowledgements

Jeonggyu Huh received financial support from the National Research Foundation of Korea (No. RS-2025-00562904).

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

# A. Anchored costate–BPTT correspondence

This appendix focuses on what backpropagation through time (BPTT) computes when differentiating anchored Monte Carlo rollouts, and it proves the costate–BPTT bridge used in Theorem 2.1. All statements here are formulated for a generic bounded continuous kernel $D(t_0, t)$ (two-time discount), covering: (i) multiplicative but time-inhomogeneous survival discounting (Case 1), (ii) time-homogeneous but non-multiplicative hyperbolic discounting (Case 2), and (iii) time-varying hyperbolic discounting (Case 3).

For each fixed anchor $t_0$, BPTT yields an anchored closed-loop sensitivity recursion. In the multiplicative regime (Case 1), this anchored recursion is consistent with the classical anchored PMP adjoint (up to the usual envelope conditions). When multiplicativity fails (Cases 2–3), the same anchored recursion still gives the natural costate proxy, and its diagonal specialization $t_0 = t_k$ is the quantity later used by Stage 2.

## A.1. Setup: anchored objective, Hamiltonian, and assumptions

We consider the controlled diffusion on $[0, T]$:

$$dS_t = b(t, S_t, u_t)\, dt + \sigma(t, S_t, u_t)\, dW_t, \qquad S_{t_0} = s_0,$$

with an admissible action set $\mathcal{U}(s)$ and a (possibly non-multiplicative) discount kernel $D(t_0, t)$. For a fixed anchor $(t_0, s_0)$, the anchored objective is

$$J(t_0, s_0; u) := \mathbb{E}\left[\int_{t_0}^{T} D(t_0, t)\, \ell(t, S_t, u_t)\, dt + D(t_0, T)\, g(S_T)\right].$$

Define the anchored Hamiltonian

$$H(t_0, t, s, u, y, z) := D(t_0, t)\, \ell(t, s, u) + \langle y, b(t, s, u)\rangle + \mathrm{Tr}\big(z^\top \sigma(t, s, u)\big). \tag{23}$$

**Standing assumptions (for discrete-to-continuous statements).**   We assume:

1. (**Regularity of $b, \sigma$.**)  $b, \sigma$ are globally Lipschitz in $s$ with linear growth, and $\sigma\sigma^\top$ is uniformly non-degenerate. Moreover, $b, \sigma$ are $C^1$ in $(s, u)$ with derivatives $\partial_s b, \partial_u b, \partial_s \sigma, \partial_u \sigma$ that are globally Lipschitz (or at least of polynomial growth ensuring the $L^2$ expansions below).

2. (**Regularity of costs.**) $\ell(\cdot, \cdot, \cdot)$ and $g(\cdot)$ are $C^1$ in $s$ (and differentiable in $u$ when needed) with polynomial growth.

3. (**Discount kernel.**) $D(t_0, t)$ is bounded and continuous on $\{(t_0, t) : 0 \leq t_0 \leq t \leq T\}$ with $D(t, t) = 1$.

4. (**Differentiable policy class.**) The policy class is differentiable in state: $u_\theta(t, s)$ is $C^1$ in $s$ with square-integrable Jacobians, and $u_\theta(t, s) \in \mathcal{U}(s)$.

## A.2. Discrete rollouts and the exact closed-loop BPTT recursion

Fix an anchor $(t_0, s_0)$ and discretize $[t_0, T]$ by $t_k = t_0 + k\Delta t$, $k = 0, \ldots, N$. Consider the Euler–Maruyama rollout under a differentiable feedback policy $u_\theta(t, s)$:

$$S_{k+1} = S_k + b(t_k, S_k, u_k)\, \Delta t + \sigma(t_k, S_k, u_k)\, \Delta W_k, \qquad u_k := u_\theta(t_k, S_k). \tag{24}$$

Define the discrete anchored return

$$\widehat{J}(t_0, s_0; \theta) := \sum_{k=0}^{N-1} D(t_0, t_k)\, \ell(t_k, S_k, u_k)\, \Delta t + D(t_0, T)\, g(S_N). \tag{25}$$

Let the *pathwise BPTT costates* be the state sensitivities

$$\lambda_k^{\mathrm{pw}} := \frac{\partial \widehat{J}(t_0, s_0; \theta)}{\partial S_k}, \qquad k = 0, \ldots, N.$$

**One-step notation.** Define the one-step reward and transition map by

$$r_k := D(t_0, t_k) \, \ell(t_k, S_k, u_k) \, \Delta t, \qquad u_k := u_\theta(t_k, S_k),$$

$$S_{k+1} := F_k(S_k, u_k, \Delta W_k),$$

where $F_k$ is the Euler update in (24).

**Proposition A.1** (Exact BPTT recursion (closed-loop adjoint with a policy-Jacobian term)). *The pathwise BPTT costates satisfy*

$$\lambda_N^{\mathrm{pw}} = D(t_0, T) \, \nabla g(S_N),$$
$$\lambda_k^{\mathrm{pw}} = \partial_{S_k} r_k + (\partial_{S_k} F_k)^\top \lambda_{k+1}^{\mathrm{pw}} + (\partial_{S_k} u_k)^\top G_k, \tag{26}$$

*for $k = 0, \dots, N-1$, where*

$$G_k := \partial_{u_k} r_k + (\partial_{u_k} F_k)^\top \lambda_{k+1}^{\mathrm{pw}}. \tag{27}$$

*Proof.* Fix a single Monte Carlo rollout, i.e., condition on a realization of the noise increments $\{\Delta W_k\}_{k=0}^{N-1}$. Then the rollout induces a deterministic computation graph with the closed-loop dependence $u_k = u_\theta(t_k, S_k)$ and the one-step transition $S_{k+1} = F_k(S_k, u_k, \Delta W_k)$. For $k = 0, \dots, N$, define the *tail return* from time index $k$:

$$\widehat{J}_k := \sum_{i=k}^{N-1} r_i + D(t_0, T) \, g(S_N), \qquad \text{so that} \qquad \widehat{J}_0 = \widehat{J}(t_0, s_0; \theta).$$

By construction, $\widehat{J}_k$ depends on $S_k$ only through the subgraph $S_k \to (u_k, r_k, S_{k+1}) \to \cdots \to S_N$, hence

$$\lambda_k^{\mathrm{pw}} = \frac{\partial \widehat{J}(t_0, s_0; \theta)}{\partial S_k} = \frac{\partial \widehat{J}_k}{\partial S_k}.$$

**Terminal condition.** At $k = N$, $\widehat{J}_N = D(t_0, T) \, g(S_N)$, hence

$$\lambda_N^{\mathrm{pw}} = \frac{\partial \widehat{J}_N}{\partial S_N} = D(t_0, T) \, \nabla g(S_N),$$

which proves the terminal condition in (26).

**Backward recursion.** For $k = 0, \dots, N-1$, the tail return satisfies

$$\widehat{J}_k = r_k + \widehat{J}_{k+1}, \qquad \text{with} \qquad S_{k+1} = F_k(S_k, u_k, \Delta W_k), \ \ u_k = u_\theta(t_k, S_k).$$

Differentiate both sides with respect to $S_k$ and use the chain rule:

$$\lambda_k^{\mathrm{pw}} = \frac{\partial \widehat{J}_k}{\partial S_k} = \frac{\partial r_k}{\partial S_k} + \left( \frac{\partial S_{k+1}}{\partial S_k} \right)^\top \frac{\partial \widehat{J}_{k+1}}{\partial S_{k+1}}$$
$$= \frac{\partial r_k}{\partial S_k} + \left( \frac{\partial S_{k+1}}{\partial S_k} \right)^\top \lambda_{k+1}^{\mathrm{pw}}.$$

Expand the two Jacobians under the closed-loop dependence $u_k = u_\theta(t_k, S_k)$.

*(i) Reward term.* Viewing $r_k = r_k(S_k, u_k)$,

$$\frac{\partial r_k}{\partial S_k} = \partial_{S_k} r_k + (\partial_{S_k} u_k)^\top \partial_{u_k} r_k.$$

*(ii) Transition term.* Viewing $S_{k+1} = F_k(S_k, u_k, \Delta W_k)$ (with $\Delta W_k$ fixed),

$$\frac{\partial S_{k+1}}{\partial S_k} = \partial_{S_k} F_k + \partial_{u_k} F_k \, \partial_{S_k} u_k.$$

Substituting (i) and (ii) gives

$$\lambda_k^{\mathrm{pw}} = \partial_{S_k} r_k + (\partial_{S_k} u_k)^\top \partial_{u_k} r_k + \left( \partial_{S_k} F_k + \partial_{u_k} F_k \, \partial_{S_k} u_k \right)^\top \lambda_{k+1}^{\mathrm{pw}}$$

$$= \partial_{S_k} r_k + (\partial_{S_k} F_k)^\top \lambda_{k+1}^{\mathrm{pw}} + (\partial_{S_k} u_k)^\top \left( \partial_{u_k} r_k + (\partial_{u_k} F_k)^\top \lambda_{k+1}^{\mathrm{pw}} \right),$$

and defining $G_k$ as in (27) completes the proof. $\qquad\square$

**Interpretation.** The additional term $(\partial_{S_k} u_k)^\top G_k$ is the standard policy-Jacobian contribution in closed-loop differentiation. The quantity $G_k$ is the discrete analogue of a Hamiltonian stationarity residual: it vanishes at an interior Pontryagin-stationary point (discrete analogue of $\partial_u H = 0$).

**Corollary A.2** (Envelope cancellation at Pontryagin-stationary points)**.** *If $G_k = 0$ along the rollout (e.g., at an interior maximizer in action space), then the BPTT recursion reduces to the standard discrete adjoint recursion without the policy-Jacobian term:*

$$\lambda_k^{\mathrm{pw}} = \partial_{S_k} r_k + (\partial_{S_k} F_k)^\top \lambda_{k+1}^{\mathrm{pw}}.$$

## A.3. Adapted projections and a BSDE limit (diagonal/anchored adjoint)

BPTT produces *pathwise* costates $\lambda_k^{\mathrm{pw}}$. For Pontryagin-type synthesis we use their predictable (adapted) projection and the associated martingale coefficient. Define the one-step predictable projections

$$\lambda_k := \mathbb{E}[\lambda_k^{\mathrm{pw}} \mid \mathcal{F}_{t_k}], \qquad Z_k := \frac{1}{\Delta t} \mathbb{E}\left[ \lambda_{k+1}^{\mathrm{pw}} (\Delta W_k)^\top \mid \mathcal{F}_{t_k} \right], \tag{28}$$

where $\Delta W_k := W_{t_{k+1}} - W_{t_k}$ and $Z_k \in \mathbb{R}^{d \times q}$.

**Proposition A.3** (Continuous-time limit: closed-loop adjoint BSDE (anchored at $t_0$))**.** *Under the standing assumptions, as $\Delta t \to 0$ the piecewise-constant interpolations of $(\lambda_k, Z_k)$ converge in $L^2$ (along a refining sequence of grids) to an adapted pair $(\lambda_t, Z_t)$ that satisfies a closed-loop adjoint BSDE for the anchored objective $J(t_0, s_0; u_\theta)$:*

$$d\lambda_t = -\Big( \partial_s H(t_0, t, S_t, u_\theta(t, S_t), \lambda_t, Z_t) + (\partial_s u_\theta(t, S_t))^\top \partial_u H(t_0, t, S_t, u_\theta(t, S_t), \lambda_t, Z_t) \Big) \, dt$$

$$\qquad + Z_t \, dW_t, \tag{29}$$

$$\lambda_T = D(t_0, T) \nabla g(S_T).$$

*Moreover, if the diagonal Pontryagin stationarity holds (interior case) so that $\partial_u H(t_0, t, S_t, u_\theta(t, S_t), \lambda_t, Z_t) = 0$ a.s. for a.e. $t$, then (29) reduces to the standard anchored adjoint BSDE (without the policy-Jacobian term), consistent with the classical PMP (Pontryagin et al., 1962; Yong & Zhou, 1999).*

*Proof.* We work on a fixed anchor $(t_0, s_0)$ and a refining sequence of uniform grids $t_k = t_0 + k\Delta t$, $\Delta t = (T - t_0)/N$. For clarity write $u_k := u_\theta(t_k, S_k)$, $b_k := b(t_k, S_k, u_k)$ and $\sigma_k := \sigma(t_k, S_k, u_k)$.

**Step 1: recall the closed-loop BPTT recursion.** From Proposition A.1,

$$\lambda_k^{\mathrm{pw}} = \partial_{S_k} r_k + (\partial_{S_k} F_k)^\top \lambda_{k+1}^{\mathrm{pw}} + (\partial_{S_k} u_k)^\top G_k, \qquad k = 0, \ldots, N-1, \tag{30}$$

with terminal condition $\lambda_N^{\mathrm{pw}} = D(t_0, T) \nabla g(S_N)$. Here $r_k = D(t_0, t_k) \ell(t_k, S_k, u_k) \Delta t$ and the Euler step is

$$F_k(s, u, \Delta W) = s + b(t_k, s, u) \Delta t + \sigma(t_k, s, u) \Delta W.$$

By construction, $\partial_{S_k} F_k$ and $\partial_{u_k} F_k$ denote *partial* derivatives of $F_k$ with respect to its first and second arguments.

**Step 2: define the predictable projections and the conditional $L^2$ decomposition.** Let $\mathcal{F}_{t_k}$ be the sigma-field generated by the Brownian path up to $t_k$. Define

$$\lambda_k := \mathbb{E}[\lambda_k^{\mathrm{pw}} \mid \mathcal{F}_{t_k}], \qquad \tilde{\lambda}_{k+1} := \mathbb{E}[\lambda_{k+1}^{\mathrm{pw}} \mid \mathcal{F}_{t_k}] = \mathbb{E}[\lambda_{k+1} \mid \mathcal{F}_{t_k}],$$

and define $Z_k$ as in (28). Then the conditional $L^2$ projection yields

$$\lambda_{k+1}^{\mathrm{pw}} = \tilde{\lambda}_{k+1} + Z_k \, \Delta W_k + R_k, \qquad \mathbb{E}[R_k \mid \mathcal{F}_{t_k}] = 0, \quad \mathbb{E}[R_k (\Delta W_k)^\top \mid \mathcal{F}_{t_k}] = 0. \tag{31}$$

**Step 3: take conditional expectation of** (30). Taking $\mathbb{E}[\cdot \mid \mathcal{F}_{t_k}]$ in (30) gives

$$\lambda_k = \mathbb{E}[\partial_{S_k} r_k \mid \mathcal{F}_{t_k}] + \mathbb{E}[(\partial_{S_k} F_k)^\top \lambda_{k+1}^{\mathrm{pw}} \mid \mathcal{F}_{t_k}] + \mathbb{E}[(\partial_{S_k} u_k)^\top G_k \mid \mathcal{F}_{t_k}]. \tag{32}$$

Since $\partial_{S_k} u_k = \partial_s u_\theta(t_k, S_k)$ is $\mathcal{F}_{t_k}$-measurable, we will use

$$\mathbb{E}[(\partial_{S_k} u_k)^\top G_k \mid \mathcal{F}_{t_k}] = (\partial_s u_\theta(t_k, S_k))^\top \mathbb{E}[G_k \mid \mathcal{F}_{t_k}].$$

**Step 4: expand** $\mathbb{E}[(\partial_{S_k} F_k)^\top \lambda_{k+1}^{\mathrm{pw}} \mid \mathcal{F}_{t_k}]$. From the Euler map,

$$\partial_{S_k} F_k = I_d + \partial_s b(t_k, S_k, u_k) \, \Delta t + \sum_{\ell=1}^q \partial_s \sigma^{(\ell)}(t_k, S_k, u_k) \, \Delta W_k^{(\ell)},$$

where $\sigma^{(\ell)}$ is the $\ell$-th column of $\sigma$. Insert (31) and use conditional moments $\mathbb{E}[\Delta W_k \mid \mathcal{F}_{t_k}] = 0$ and $\mathbb{E}[\Delta W_k (\Delta W_k)^\top \mid \mathcal{F}_{t_k}] = \Delta t \, I_q$. A direct computation yields

$$\mathbb{E}[(\partial_{S_k} F_k)^\top \lambda_{k+1}^{\mathrm{pw}} \mid \mathcal{F}_{t_k}] = \tilde{\lambda}_{k+1} + (\partial_s b(t_k, S_k, u_k))^\top \tilde{\lambda}_{k+1} \, \Delta t$$
$$+ \sum_{\ell=1}^q (\partial_s \sigma^{(\ell)}(t_k, S_k, u_k))^\top Z_k^{(\ell)} \, \Delta t + o_{L^2}(\Delta t), \tag{33}$$

where $Z_k^{(\ell)}$ denotes the $\ell$-th column of $Z_k$, and $o_{L^2}(\Delta t)$ collects higher-order Euler remainders and terms controlled under the standing assumptions.

**Step 5: expand** $\mathbb{E}[G_k \mid \mathcal{F}_{t_k}]$. Recall

$$G_k = \partial_{u_k} r_k + (\partial_{u_k} F_k)^\top \lambda_{k+1}^{\mathrm{pw}}, \qquad \partial_{u_k} F_k = \partial_u b(t_k, S_k, u_k) \, \Delta t + \sum_{\ell=1}^q \partial_u \sigma^{(\ell)}(t_k, S_k, u_k) \, \Delta W_k^{(\ell)}.$$

Using (31) and conditional moments of $\Delta W_k$,

$$\mathbb{E}[G_k \mid \mathcal{F}_{t_k}] = \partial_{u_k} r_k + (\partial_u b(t_k, S_k, u_k))^\top \tilde{\lambda}_{k+1} \, \Delta t$$
$$+ \sum_{\ell=1}^q (\partial_u \sigma^{(\ell)}(t_k, S_k, u_k))^\top Z_k^{(\ell)} \, \Delta t + o_{L^2}(\Delta t). \tag{34}$$

**Step 6: collect terms and match the Hamiltonian derivatives.** First,

$$\partial_{S_k} r_k = D(t_0, t_k) \, \partial_s \ell(t_k, S_k, u_k) \, \Delta t, \qquad \partial_{u_k} r_k = D(t_0, t_k) \, \partial_u \ell(t_k, S_k, u_k) \, \Delta t.$$

Plug (33) and (34) into (32) to obtain

$$\lambda_k = \tilde{\lambda}_{k+1} + \Big( \partial_s H(t_0, t_k, S_k, u_k, \tilde{\lambda}_{k+1}, Z_k) + (\partial_s u_\theta(t_k, S_k))^\top \partial_u H(t_0, t_k, S_k, u_k, \tilde{\lambda}_{k+1}, Z_k) \Big) \Delta t + o_{L^2}(\Delta t), \tag{35}$$

where $H$ is the anchored Hamiltonian (23) and the identities follow by direct differentiation of $H$ with respect to $s$ and $u$.

**Step 7: continuous-time limit.** Let $\bar{\lambda}^{\Delta t}$ and $\bar{Z}^{\Delta t}$ be the piecewise-constant interpolations on $[t_0, T]$ defined by $\bar{\lambda}_t^{\Delta t} := \lambda_k$ and $\bar{Z}_t^{\Delta t} := Z_k$ for $t \in [t_k, t_{k+1})$. Under the standing assumptions, Euler–Maruyama satisfies strong $L^2$ convergence, and the backward scheme (35) is stable in $L^2$. Therefore, along a refining sequence $\Delta t \downarrow 0$, the interpolations converge in $L^2$ to an adapted pair $(\lambda_t, Z_t)$ satisfying

$$d\lambda_t = -\Big( \partial_s H(t_0, t, S_t, u_\theta(t, S_t), \lambda_t, Z_t) + (\partial_s u_\theta(t, S_t))^\top \partial_u H(t_0, t, S_t, u_\theta(t, S_t), \lambda_t, Z_t) \Big) dt + Z_t \, dW_t,$$

with terminal condition $\lambda_T = D(t_0, T)\nabla g(S_T)$, i.e., (29). (See, e.g., BSDE stability/discretization results in Yong & Zhou (1999) and references therein.)

**Step 8: envelope cancellation under Pontryagin stationarity.** If the (interior) Pontryagin stationarity condition holds so that $\partial_u H(t_0, t, S_t, u_\theta(t, S_t), \lambda_t, Z_t) = 0$ a.s. for a.e. $t$, then the policy-Jacobian drift term vanishes and (29) reduces to the standard anchored adjoint BSDE of the Pontryagin maximum principle (Pontryagin et al., 1962; Yong & Zhou, 1999). $\square$

## A.4. A theorem-level bridge: anchored costate–BPTT correspondence

The previous propositions already contain the key ingredients for the warm-up-policy interpretation of BPTT: Proposition A.1 gives the exact closed-loop state-sensitivity recursion, while Proposition A.3 identifies its predictable projection with the anchored adjoint BSDE modulo the policy-Jacobian correction. The next theorem packages these facts in the same form used by the delay paper: the comparison is with the *exact anchored adjoint of the warm-up policy*, not with an unknown optimal costate.

**Theorem A.4** ((restatement of Theorem 2.1) Anchored costate–BPTT correspondence). *Fix an anchor $t_0$ and the warm-up policy $u_{\theta^\star}$. Let $S_k$ be the Euler rollout, let $u_k := u_{\theta^\star}(t_k, S_k)$, and define*

$$\lambda_k^{\mathrm{pw}} := \frac{\partial \widehat{J}(t_0, s_0; \theta)}{\partial S_k}, \qquad \lambda_k := \mathbb{E}[\lambda_k^{\mathrm{pw}} \mid \mathcal{F}_{t_k}], \qquad \lambda_{k+1|k} := \mathbb{E}[\lambda_{k+1}^{\mathrm{pw}} \mid \mathcal{F}_{t_k}], \qquad Z_k := \frac{1}{\Delta t}\, \mathbb{E}[\lambda_{k+1}^{\mathrm{pw}}(\Delta W_k)^\top \mid \mathcal{F}_{t_k}].$$

*For compactness, also set*

$$H_k^{t_0}(s, u; p, Z) := D(t_0, t_k)\, \ell(t_k, s, u) + p^\top b(t_k, s, u) + \langle Z, \sigma(t_k, s, u) \rangle_F,$$

$$D_k^s := \partial_s H_k^{t_0}(S_k, u_k; \lambda_{k+1|k}, Z_k), \qquad r_k^{\mathrm{FOC}} := \partial_u H_k^{t_0}(S_k, u_k; \lambda_{k+1|k}, Z_k), \qquad \mathcal{C}_k^{\mathrm{FOC}} := \big(r_k^{\mathrm{FOC}}\big)^\top \partial_s u_{\theta^\star}(t_k, S_k).$$

*Then there exists an $\mathcal{F}_{t_k}$-measurable remainder $\rho_k^{\mathrm{disc}}$ with $\|\rho_k^{\mathrm{disc}}\|_{L^2} = o(\Delta t)$ such that*

$$\lambda_k = \lambda_{k+1|k} + \big(D_k^s + \mathcal{C}_k^{\mathrm{FOC}}\big)\Delta t + \rho_k^{\mathrm{disc}}.$$

*Moreover, if $\|\partial_s u_{\theta^\star}(t_k, S_k)\|_{L^\infty} \leq C_u$ and $\|r_k^{\mathrm{FOC}}\|_{L^2} \leq \varepsilon_k^{\mathrm{FOC}}$, then*

$$\|\mathcal{C}_k^{\mathrm{FOC}}\|_{L^2} \leq C_u\, \varepsilon_k^{\mathrm{FOC}}, \qquad \|\lambda_k - \lambda_{k+1|k} - D_k^s \Delta t\|_{L^2} \leq C_u\, \varepsilon_k^{\mathrm{FOC}}\, \Delta t + o(\Delta t).$$

*In particular, exact predictable stationarity $r_k^{\mathrm{FOC}} = 0$ removes the policy-sensitivity mismatch, leaving only the standard Euler discretization error. Specializing the anchor to the decision time, $t_0 = t_k$, gives the diagonal costate proxy used by Stage 2.*

*Proof.* The drift identity is exactly the discrete scheme (35) from Proposition A.3, after setting

$$\rho_k^{\mathrm{disc}} := \lambda_k - \lambda_{k+1|k} - \big(D_k^s + \mathcal{C}_k^{\mathrm{FOC}}\big)\Delta t.$$

By Proposition A.3, this remainder satisfies $\|\rho_k^{\mathrm{disc}}\|_{L^2} = o(\Delta t)$.

For the residual estimate, Hölder's inequality gives

$$\|\mathcal{C}_k^{\mathrm{FOC}}\|_{L^2} = \|(r_k^{\mathrm{FOC}})^\top \partial_s u_{\theta^\star}(t_k, S_k)\|_{L^2} \leq \|\partial_s u_{\theta^\star}(t_k, S_k)\|_{L^\infty} \|r_k^{\mathrm{FOC}}\|_{L^2} \leq C_u\, \varepsilon_k^{\mathrm{FOC}}.$$

Insert this bound into the drift identity and use the definition of $\rho_k^{\mathrm{disc}}$ to obtain

$$\|\lambda_k - \lambda_{k+1|k} - D_k^s \Delta t\|_{L^2} \leq \|\mathcal{C}_k^{\mathrm{FOC}}\|_{L^2}\, \Delta t + \|\rho_k^{\mathrm{disc}}\|_{L^2} \leq C_u\, \varepsilon_k^{\mathrm{FOC}}\, \Delta t + o(\Delta t).$$

The final statement follows immediately when $r_k^{\mathrm{FOC}} = 0$. $\square$

**Lemma A.5** (Canonical one-step martingale representation). *Under the notation of Theorem A.4, if*

$$R_k := \lambda_{k+1}^{\mathrm{pw}} - \lambda_{k+1|k} - Z_k \Delta W_k,$$

*then $\mathbb{E}[R_k \mid \mathcal{F}_{t_k}] = 0$, $\mathbb{E}[R_k(\Delta W_k)^\top \mid \mathcal{F}_{t_k}] = 0$, and*

$$\lambda_{k+1}^{\mathrm{pw}} = \lambda_k - \big(D_k^s + \mathcal{C}_k^{\mathrm{FOC}}\big)\Delta t - \rho_k^{\mathrm{disc}} + Z_k \Delta W_k + R_k.$$

*Proof.* Use the conditional decomposition already introduced in (31):

$$\lambda_{k+1}^{\mathrm{pw}} = \lambda_{k+1|k} + Z_k \Delta W_k + R_k, \qquad \mathbb{E}[R_k \mid \mathcal{F}_{t_k}] = 0, \quad \mathbb{E}[R_k (\Delta W_k)^\top \mid \mathcal{F}_{t_k}] = 0.$$

Substituting the drift identity from Theorem A.4,

$$\lambda_{k+1|k} = \lambda_k - \left( D_k^s + \mathcal{C}_k^{\mathrm{FOC}} \right) \Delta t - \rho_k^{\mathrm{disc}},$$

yields the claimed one-step form. $\qquad\qquad\qquad\qquad\qquad\qquad\qquad\qquad\qquad\qquad\qquad\qquad\qquad\square$

## B. Stage 2 diagonal Pontryagin projection and near-equilibrium guarantee

This appendix focuses on what Stage 2 explicitly enforces and on the proof of Theorem 2.2. We use the Hamiltonian $H$ and the standing assumptions introduced in Appendix A. The argument has two parts: first, Proposition B.1 shows that the Stage 2 plug-in solve enforces the *diagonal* Pontryagin condition with respect to the anchored-at-the-decision-time continuation problem; second, the short-slab estimates below transfer this enforcement to the exact diagonal control, yielding a near-optimality or near-equilibrium guarantee depending on whether the discount kernel is multiplicative.

### B.1. What Stage 2 enforces: diagonal Pontryagin projection (time-consistency restoration)

Stage 2 is designed to enforce a *diagonal* Pontryagin condition (anchor = decision time). At a query point $(t, s)$, we construct Monte Carlo rollouts anchored at time $t$ and define anchored returns $\widehat{J}^{(j)}(t, s; \theta^\star)$. BPTT state-gradients yield pathwise diagonal costate estimates

$$\lambda^{(j)}(t, s) := \frac{\partial \widehat{J}^{(j)}(t, s; \theta^\star)}{\partial s}, \qquad \widehat{\lambda}(t, s) := \frac{1}{M_{\mathrm{MC}}} \sum_{j=1}^{M_{\mathrm{MC}}} \lambda^{(j)}(t, s).$$

Optionally, when $\sigma$ depends on $u$, we also estimate the diagonal martingale coefficient $\widehat{Z}(t, s)$ by one-step $L^2$ regression (cf. (28)). We then synthesize the action by diagonal Hamiltonian maximization (or a constrained Newton/log-barrier solve):

$$u^{\mathrm{proj}}(t, s) \in \arg \max_{u \in \mathcal{U}(s)} H\big(t, t, s, u, \widehat{\lambda}(t, s), \widehat{Z}(t, s)\big), \tag{36}$$

where $H$ is the anchored Hamiltonian (23).

**Proposition B.1** (Diagonal Pontryagin enforcement by Stage 2). *Assume (i) the inner action-space maximization (36) is solved exactly (or to a prescribed tolerance) and (ii) the diagonal estimators $\widehat{\lambda}(t, s)$ (and $\widehat{Z}(t, s)$ when needed) are accurate for the anchored-at-$t$ objective. Then $u^{\mathrm{proj}}(t, s)$ satisfies the diagonal Pontryagin equilibrium condition at $(t, s)$ up to solver/statistical tolerance. When $D$ is multiplicative, this diagonal condition reduces to the classical Pontryagin optimality condition. Under standard concavity assumptions, this diagonal condition is the appropriate first-order characterization of time-consistent equilibrium for non-multiplicative discounting (Ekeland & Lazrak, 2006a; Björk & Murgoci, 2014; Yong, 2012).*

*Proof.* Fix a query point $(t, s)$.

**Step 1: Stage 2 action-space problem.** By definition of Stage 2, we form Monte Carlo rollouts starting from $(t, s)$ under the frozen warm-start policy $u_{\theta^\star}$ and evaluate an objective *anchored at the decision time* $t$. From these rollouts we construct the diagonal costate estimator $\widehat{\lambda}(t, s)$ (and $\widehat{Z}(t, s)$ when needed), and then compute

$$u^{\mathrm{proj}}(t, s) \in \arg \max_{u \in \mathcal{U}(s)} H\big(t, t, s, u, \widehat{\lambda}(t, s), \widehat{Z}(t, s)\big). \tag{37}$$

Assumption (i) states that the numerical solver returns an $\varepsilon_{\mathrm{opt}}$-accurate maximizer in the sense that

$$H\big(t, t, s, u^{\mathrm{proj}}(t, s), \widehat{\lambda}(t, s), \widehat{Z}(t, s)\big) \geq \sup_{u \in \mathcal{U}(s)} H\big(t, t, s, u, \widehat{\lambda}(t, s), \widehat{Z}(t, s)\big) - \varepsilon_{\mathrm{opt}}. \tag{38}$$

Assumption (ii) states that $(\widehat{\lambda}, \widehat{Z})$ are accurate for the anchored-at-$t$ continuation problem; for instance,

$$\|\widehat{\lambda}(t, s) - \lambda(t, s)\| \leq \varepsilon_\lambda, \qquad \|\widehat{Z}(t, s) - Z(t, s)\| \leq \varepsilon_Z,$$

where $(\lambda, Z)$ denote the true (diagonal) adjoint variables for the anchored objective at $t$.

**Step 2: diagonal maximization implies (approximate) diagonal stationarity.** Consider first the unconstrained or interior case. Suppose $u^{\mathrm{proj}}(t,s)$ lies in the interior of $\mathcal{U}(s)$. If the map $u \mapsto H(t,t,s,u,\widehat{\lambda}(t,s),\widehat{Z}(t,s))$ is concave, differentiable, and has $L$-Lipschitz gradient in $u$ (local $L$-smoothness), then an exact maximizer satisfies

$$\partial_u H\big(t,t,s,u^{\mathrm{proj}}(t,s),\widehat{\lambda}(t,s),\widehat{Z}(t,s)\big) = 0. \tag{39}$$

If we solve only to tolerance (38), standard smooth concave maximization arguments yield an *approximate* stationarity statement of the form

$$\big\|\partial_u H\big(t,t,s,u^{\mathrm{proj}}(t,s),\widehat{\lambda}(t,s),\widehat{Z}(t,s)\big)\big\| \leq \sqrt{2L\,\varepsilon_{\mathrm{opt}}}. \tag{40}$$

In the constrained case $\mathcal{U}(s) = \{u : g_i(u,s) \leq 0\}$, Stage 2 is implemented by an interior-point / log-barrier or projected Newton solve, so the appropriate first-order condition is the KKT condition. Under standard constraint qualification and concavity in $u$, the maximizer satisfies

$$\partial_u H(\cdots) + \sum_i \eta_i\, \partial_u g_i(u^{\mathrm{proj}}(t,s),s) = 0, \qquad \eta_i \geq 0, \quad \eta_i g_i(u^{\mathrm{proj}}(t,s),s) = 0,$$

up to solver tolerance (and, if a log-barrier with parameter $\mu > 0$ is used, up to the usual barrier bias that vanishes as $\mu \downarrow 0$).

**Step 3: transfer from estimated to true diagonal adjoints.** Let $(\lambda(t,s), Z(t,s))$ denote the true diagonal adjoint variables for the anchored-at-$t$ continuation problem. By smoothness of $H$ in $(\lambda, Z)$, we have a stability bound

$$\big\|\partial_u H\big(t,t,s,u^{\mathrm{proj}},\lambda(t,s),Z(t,s)\big)\big\| \leq \big\|\partial_u H\big(t,t,s,u^{\mathrm{proj}},\widehat{\lambda}(t,s),\widehat{Z}(t,s)\big)\big\|$$
$$+ L_\lambda\,\|\widehat{\lambda}(t,s) - \lambda(t,s)\| + L_Z\,\|\widehat{Z}(t,s) - Z(t,s)\|, \tag{41}$$

for appropriate local Lipschitz constants $L_\lambda, L_Z$. Combining (40) with (41) yields the claimed diagonal Pontryagin optimality/equilibrium condition up to *solver* and *statistical* tolerances.

**Step 4: relation to time-consistent equilibrium under non-multiplicative discounting.** For non-multiplicative kernels, the continuation problem depends on the anchor time, and equilibrium notions in time-inconsistent control are characterized by *diagonal* (anchor = decision time) first-order conditions (extended/equilibrium HJB / extended PMP; see Ekeland & Lazrak, 2006a; Björk & Murgoci, 2014; Yong, 2012). Stage 2 enforces exactly this diagonal condition by anchoring rollouts at $t$ and maximizing the corresponding Hamiltonian. This completes the proof. □

### B.2. Short-slab stability and near-optimality / near-equilibrium of Stage 2

Proposition B.1 shows that Stage 2 enforces the diagonal Pontryagin condition with respect to the plug-in adjoint estimate. The missing quantitative step is to relate this enforcement to the *exact* diagonal Pontryagin control. The result in this section fills that gap. The structure parallels the short-slab projection theorem used in the delay setting, but there is one essential difference: in the non-multiplicative regime the natural consequence is a *near-equilibrium* guarantee rather than a global value-optimality statement.

**Theorem B.2** ((full version of Theorem 2.2) Short-slab stability and quantitative control accuracy of the diagonal Pontryagin projection). *Let $(\bar{S}_k, \bar{u}_k)$ denote the Stage 1 warm-up trajectory and control on the Euler mesh, with $\bar{u}_k := u_{\theta^\star}(t_k, \bar{S}_k)$. For each mesh time $t_k$, let $\bar{\vartheta}_k = (\bar{\lambda}_k, \bar{Z}_k)$ be the exact diagonal adjoint pair for the anchored-at-$t_k$ continuation problem under the warm-up policy, evaluated at $\bar{S}_k$, and let $\widehat{\vartheta}_k = (\widehat{\lambda}_k, \widehat{Z}_k)$ be the Stage 2 estimator. Let $u_k^\star$ denote the exact diagonal Pontryagin control, namely the classical optimal control when $D$ is multiplicative and the time-consistent equilibrium control otherwise. Partition the Euler mesh into slabs $I_\ell$ of length $\tau$ and define*

$$\|v\|_{\ell,2}^2 := \sum_{k \in I_\ell} \mathbb{E}|v_k|^2\,\Delta t, \qquad \delta_\ell^{\mathrm{diag}} := \|\widehat{\vartheta} - \bar{\vartheta}\|_{\ell,2}.$$

*Under the local projection regularity, short-slab propagation, residual-to-solution transfer, and finite patching conditions stated below, there exists $\tau_0 > 0$ such that, for every slab length $\tau \leq \tau_0$, the Stage 2 projected control $u^{\mathrm{proj}}$ satisfies*

$$\sum_{\ell=0}^{L-1} \|u^{\mathrm{proj}} - u^\star\|_{\ell,2} \leq C_{\mathrm{diag}} \sum_{\ell=0}^{L-1} \Big(\|r^{\mathrm{warm}}\|_{\ell,2} + \delta_\ell^{\mathrm{diag}} + \eta_\ell^{\mathrm{num}}\Big). \tag{42}$$

*If, moreover, the Stage 2 estimator obeys*

$$\delta_\ell^{\mathrm{diag}} \leq C_{\mathrm{disc}}\, \Delta t^{1/2} + C_{\mathrm{MC}}\, M^{-1/2},$$

*then the same bound holds with $\delta_\ell^{\mathrm{diag}}$ replaced by the right-hand side. When $D$ is multiplicative, (42) yields a value-gap bound; when $D$ is non-multiplicative, it yields an analogous bound on the diagonal equilibrium defect. When $\sigma$ is independent of $u$, the $Z$-terms can be dropped throughout.*

**Diagonal projection map, warm-up projection, and slab norms.** Fix an Euler mesh $t_k = t_0 + k\Delta t$ and write

$$H_k^{\mathrm{diag}}(s, u; \vartheta) := H(t_k, t_k, s, u, y, z), \qquad \vartheta = (y, z).$$

For the pair $\vartheta = (y, z)$ we use any fixed product norm $|\vartheta| := |y| + |z|_F$. Define the pointwise diagonal projection map

$$\mathcal{P}_k(s, \vartheta) \in \underset{u \in \mathcal{U}(s)}{\arg\max}\, H_k^{\mathrm{diag}}(s, u; \vartheta).$$

Let $(\bar{S}_k, \bar{u}_k)$ be the Stage 1 warm-up state/control pair, with $\bar{u}_k := u_{\theta^\star}(t_k, \bar{S}_k)$. For each mesh time $t_k$, let $\bar{\vartheta}_k = (\bar{\lambda}_k, \bar{Z}_k)$ denote the exact diagonal adjoint pair for the anchored-at-$t_k$ continuation problem under the warm-up policy, evaluated at $\bar{S}_k$. Let $\widehat{\vartheta}_k = (\widehat{\lambda}_k, \widehat{Z}_k)$ be the Stage 2 estimator used by the algorithm. We define the *ideal warm-up projection*

$$u_k^\dagger := \mathcal{P}_k(\bar{S}_k, \bar{\vartheta}_k).$$

Let $u_k^\star$ denote the exact diagonal Pontryagin control; equivalently, $u^\star$ is the classical optimal control in the multiplicative regime and the exact time-consistent equilibrium control in the non-multiplicative regime. Partition the Euler mesh into slabs

$$I_\ell := \{k_\ell, \dots, k_{\ell+1} - 1\}, \qquad k_{\ell+1} - k_\ell = m_\tau, \qquad \tau = m_\tau \Delta t,$$

and define

$$\|v\|_{\ell,2}^2 := \sum_{k \in I_\ell} \mathbb{E}|v_k|^2\, \Delta t, \qquad \|v\|_{\mathrm{slab},1} := \sum_{\ell=0}^{L-1} \|v\|_{\ell,2}.$$

We also write

$$\delta_\ell^{\mathrm{diag}} := \|\widehat{\vartheta} - \bar{\vartheta}\|_{\ell,2} \qquad \text{and} \qquad \Gamma_\ell^{\mathrm{in}} := \left(\mathbb{E}|S_{k_\ell}^{\mathrm{proj}} - \bar{S}_{k_\ell}|^2\right)^{1/2},$$

where $S^{\mathrm{proj}}$ denotes the state process driven by the Stage 2 feedback control.

**Assumption B.3** (Local projection regularity and short-slab state propagation). There exist constants $m_H, L_s, L_\vartheta, C_S, C_{\mathrm{prop}} > 0$ such that the following hold on a working neighborhood of the reference trajectories.

(i) (Uniform strong concavity in the control.) For every $k$, every admissible state $s$, every admissible adjoint pair $\vartheta$, and all $u_1, u_2 \in \mathcal{U}(s)$,

$$\left\langle \nabla_u H_k^{\mathrm{diag}}(s, u_1; \vartheta) - \nabla_u H_k^{\mathrm{diag}}(s, u_2; \vartheta),\, u_1 - u_2 \right\rangle \leq -m_H |u_1 - u_2|^2.$$

(ii) (Lipschitz dependence on state and adjoint.) For every $k$, every admissible $s_1, s_2$, every admissible $\vartheta_1, \vartheta_2$, and every $u$,

$$\left| \nabla_u H_k^{\mathrm{diag}}(s_1, u; \vartheta_1) - \nabla_u H_k^{\mathrm{diag}}(s_2, u; \vartheta_2) \right| \leq L_s |s_1 - s_2| + L_\vartheta |\vartheta_1 - \vartheta_2|.$$

(iii) (Short-slab propagation of state mismatch.) For any two admissible controls $u^1, u^2$ on a slab $I_\ell$,

$$\|S^{u^1} - S^{u^2}\|_{\ell,2} \leq C_S\, \Gamma_\ell^{\mathrm{in}} + C_S\, \tau^{1/2}\, \|u^1 - u^2\|_{\ell,2},$$

where $\Gamma_\ell^{\mathrm{in}} = (\mathbb{E}|S_{k_\ell}^{u^1} - S_{k_\ell}^{u^2}|^2)^{1/2}$.

(iv) (Propagation to the next slab.) For consecutive slabs,

$$\Gamma_{\ell+1}^{\mathrm{in}} \leq C_{\mathrm{prop}}\left(\Gamma_\ell^{\mathrm{in}} + \tau^{1/2}\, \|u^1 - u^2\|_{\ell,2}\right).$$

**Warm-up residual and numerical solver residual.** Define the local diagonal warm-up residual by

$$r_k^{\text{warm}} := \text{dist}\Big(0, \nabla_u H_k^{\text{diag}}(\bar{S}_k, \bar{u}_k; \bar{\vartheta}_k) + N_{\mathcal{U}(\bar{S}_k)}(\bar{u}_k)\Big), \tag{43}$$

with slab norm

$$\|r^{\text{warm}}\|_{\ell,2}^2 := \sum_{k \in I_\ell} \mathbb{E}|r_k^{\text{warm}}|^2 \, \Delta t.$$

Assume that the numerical Stage 2 routine returns $u_k^{\text{proj}}$ satisfying

$$\text{dist}\Big(0, \nabla_u H_k^{\text{diag}}(S_k^{\text{proj}}, u_k^{\text{proj}}; \widehat{\vartheta}_k) + N_{\mathcal{U}(S_k^{\text{proj}})}(u_k^{\text{proj}})\Big) \leq \eta_k^{\text{num}}, \tag{44}$$

and define

$$\eta_\ell^{\text{num}} := \left(\sum_{k \in I_\ell} \mathbb{E}|\eta_k^{\text{num}}|^2 \, \Delta t\right)^{1/2}.$$

**Lemma B.4** (Short-slab control gap around the ideal warm-up projection). *Assume Assumption B.3 and define*

$$\alpha_\tau := \frac{L_s C_S}{m_H} \tau^{1/2}.$$

*If $\alpha_\tau < 1$, then the Stage 2 projected control satisfies*

$$\|u^{\text{proj}} - u^\dagger\|_{\ell,2} \leq \frac{1}{1 - \alpha_\tau} \left[\frac{L_s C_S}{m_H} \Gamma_\ell^{\text{in}} + \frac{\alpha_\tau}{m_H} \|r^{\text{warm}}\|_{\ell,2} + \frac{L_\vartheta}{m_H} \delta_\ell^{\text{diag}} + \frac{1}{m_H} \eta_\ell^{\text{num}}\right]. \tag{45}$$

*In particular, on the first slab (or whenever $\Gamma_\ell^{\text{in}} = 0$),*

$$\|u^{\text{proj}} - u^\dagger\|_{\ell,2} \leq \frac{1}{m_H(1 - \alpha_\tau)} \left[\alpha_\tau \|r^{\text{warm}}\|_{\ell,2} + L_\vartheta \, \delta_\ell^{\text{diag}} + \eta_\ell^{\text{num}}\right]. \tag{46}$$

*Proof.* Let

$$u_k^\sharp := \mathcal{P}_k(S_k^{\text{proj}}, \widehat{\vartheta}_k)$$

be the exact pointwise maximizer associated with the estimated diagonal adjoint. By strong concavity and the residual bound (44),

$$\|u^{\text{proj}} - u^\sharp\|_{\ell,2} \leq \frac{1}{m_H} \eta_\ell^{\text{num}}. \tag{47}$$

The same strong-concavity inverse estimate applied to the warm-up pair $(\bar{S}_k, \bar{\vartheta}_k)$ gives

$$\|u^\dagger - \bar{u}\|_{\ell,2} \leq \frac{1}{m_H} \|r^{\text{warm}}\|_{\ell,2}. \tag{48}$$

Moreover, by the pointwise Lipschitz stability of the projection map implied by Assumption B.3(i)–(ii),

$$\|u^\sharp - u^\dagger\|_{\ell,2} \leq \frac{L_s}{m_H} \|S^{\text{proj}} - \bar{S}\|_{\ell,2} + \frac{L_\vartheta}{m_H} \delta_\ell^{\text{diag}}.$$

Combining this with (47) yields

$$\|u^{\text{proj}} - u^\dagger\|_{\ell,2} \leq \frac{1}{m_H} \eta_\ell^{\text{num}} + \frac{L_s}{m_H} \|S^{\text{proj}} - \bar{S}\|_{\ell,2} + \frac{L_\vartheta}{m_H} \delta_\ell^{\text{diag}}. \tag{49}$$

Now use Assumption B.3(iii):

$$\|S^{\text{proj}} - \bar{S}\|_{\ell,2} \leq C_S \Gamma_\ell^{\text{in}} + C_S \tau^{1/2} \|u^{\text{proj}} - \bar{u}\|_{\ell,2}.$$

By the triangle inequality and (48),

$$\|u^{\text{proj}} - \bar{u}\|_{\ell,2} \leq \|u^{\text{proj}} - u^\dagger\|_{\ell,2} + \|u^\dagger - \bar{u}\|_{\ell,2} \leq \|u^{\text{proj}} - u^\dagger\|_{\ell,2} + \frac{1}{m_H} \|r^{\text{warm}}\|_{\ell,2}.$$

Insert the last two displays into (49) and absorb the resulting $\alpha_\tau \|u^{\text{proj}} - u^\dagger\|_{\ell,2}$ term onto the left-hand side. This gives (45); (46) follows immediately when $\Gamma_\ell^{\text{in}} = 0$. $\qquad\square$

**Corollary B.5** (Global slab-wise patching around the ideal warm-up projection)**.** *Assume the hypotheses of Lemma B.4 on each slab $I_\ell$ and the propagation estimate in Assumption B.3(iv). Then there exists a constant $C_{\mathrm{warm}} > 0$, depending only on $T, \tau, m_H, L_s, L_\vartheta, C_S, C_{\mathrm{prop}}$, such that*

$$\sum_{\ell=0}^{L-1} \|u^{\mathrm{proj}} - u^\dagger\|_{\ell,2} \leq C_{\mathrm{warm}} \sum_{\ell=0}^{L-1} \left( \|r^{\mathrm{warm}}\|_{\ell,2} + \delta_\ell^{\mathrm{diag}} + \eta_\ell^{\mathrm{num}} \right). \tag{50}$$

*Proof.* Iterate the local estimate (45) over slabs and use Assumption B.3(iv) to control the incoming state mismatch recursively. Because $\alpha_\tau < 1$, the resulting discrete Gronwall argument closes and yields (50). $\square$

**Assumption B.6** (Residual-to-solution transfer around the diagonal control)**.** There exists a constant $C_{\mathrm{trans}} > 0$ such that the ideal warm-up projection and the exact diagonal Pontryagin control satisfy

$$\sum_{\ell=0}^{L-1} \|u^\dagger - u^\star\|_{\ell,2} \leq C_{\mathrm{trans}} \sum_{\ell=0}^{L-1} \|r^{\mathrm{warm}}\|_{\ell,2}. \tag{51}$$

In the multiplicative regime $u^\star$ is the exact optimal control. In the non-multiplicative regime $u^\star$ is the exact diagonal equilibrium control.

*Proof of Theorem B.2.* By the triangle inequality,

$$\sum_{\ell=0}^{L-1} \|u^{\mathrm{proj}} - u^\star\|_{\ell,2} \leq \sum_{\ell=0}^{L-1} \|u^{\mathrm{proj}} - u^\dagger\|_{\ell,2} + \sum_{\ell=0}^{L-1} \|u^\dagger - u^\star\|_{\ell,2}.$$

The first term is bounded by Corollary B.5; the second term is bounded by Assumption B.6. This proves (42) with

$$C_{\mathrm{diag}} := C_{\mathrm{warm}} + C_{\mathrm{trans}}.$$

If the diagonal adjoint estimator satisfies

$$\delta_\ell^{\mathrm{diag}} \leq C_{\mathrm{disc}}\, \Delta t^{1/2} + C_{\mathrm{MC}}\, M^{-1/2},$$

then substituting this bound into (42) gives the stated rate form.

If $D$ is multiplicative, then $u^\star$ is the exact optimal control for the anchored objective, so any local Lipschitz bound of the form

$$|J(t_0, s_0; u^1) - J(t_0, s_0; u^2)| \leq L_J \sum_{\ell=0}^{L-1} \|u^1 - u^2\|_{\ell,2}$$

immediately yields the value-gap estimate

$$0 \leq J(t_0, s_0; u^\star) - J(t_0, s_0; u^{\mathrm{proj}}) \leq L_J C_{\mathrm{diag}} \sum_{\ell=0}^{L-1} \left( \|r^{\mathrm{warm}}\|_{\ell,2} + \delta_\ell^{\mathrm{diag}} + \eta_\ell^{\mathrm{num}} \right). \tag{52}$$

If $D$ is non-multiplicative, define the diagonal equilibrium defect by

$$\mathcal{E}_{\mathrm{diag}}(u) := \sum_{\ell=0}^{L-1} \|r^{\mathrm{diag}}(u)\|_{\ell,2},$$

where

$$r_k^{\mathrm{diag}}(u) := \mathrm{dist}\left( 0, \nabla_u H_k^{\mathrm{diag}}(S_k^u, u_k; \vartheta_k^u) + N_{\mathcal{U}(S_k^u)}(u_k) \right)$$

and $\vartheta_k^u = (\lambda_k^u, Z_k^u)$ is the exact diagonal adjoint pair associated with the control $u$. If the residual map $u \mapsto r^{\mathrm{diag}}(u)$ is locally Lipschitz around $u^\star$, then since $r^{\mathrm{diag}}(u^\star) = 0$ we obtain

$$\mathcal{E}_{\mathrm{diag}}(u^{\mathrm{proj}}) \leq C_{\mathrm{eq}} C_{\mathrm{diag}} \sum_{\ell=0}^{L-1} \left( \|r^{\mathrm{warm}}\|_{\ell,2} + \delta_\ell^{\mathrm{diag}} + \eta_\ell^{\mathrm{num}} \right) \tag{53}$$

for a suitable local Lipschitz constant $C_{\mathrm{eq}}$. This is the natural near-equilibrium counterpart of the value-gap estimate in the non-multiplicative regime. $\square$

Theorem B.2 immediately implies the compressed main-text statement in Theorem 2.2.

## C. Stage 1 random-anchor surrogate and time-consistency remark

Stage 1 in PG-DPO maximizes the random-anchor surrogate

$$J^{\mathrm{sur}}(\theta) := \mathbb{E}_{(t_0, s_0) \sim \nu}\big[\widehat{J}(t_0, s_0; \theta)\big], \tag{54}$$

via BPTT through rollouts.

**Proposition C.1** (Unbiased random-anchor gradient estimator). *Sampling $(t_0, s_0) \sim \nu$ and backpropagating $\widehat{J}(t_0, s_0; \theta)$ yields an unbiased estimator of $\nabla_\theta J^{\mathrm{sur}}(\theta)$.*

*Proof.* Immediate from linearity of expectation and i.i.d. sampling of anchors. $\square$

*Remark* C.2 (Why Stage 1 alone does not imply time-consistent equilibrium (non-multiplicative $D$)). When discounting is non-multiplicative, time-consistent equilibrium conditions are *diagonal* conditions with anchor = decision time (extended HJB / equilibrium control; see Ekeland & Lazrak, 2006a; Björk & Murgoci, 2014; Yong, 2012). Stationarity of the *averaged* surrogate (54) generally enforces a weighted mixture of anchor-dependent stationarity residuals and does not, in general, imply the diagonal equilibrium condition. Accordingly, we use Stage 1 as a robust warm-start that produces a stable differentiable rollout policy, while equilibrium/optimality is enforced by Stage 2.

## D. Baseline Definitions and Interpretation Guide

We compare PG-DPO against baselines that differ in their required level of model information and in how they use optimality structure. The purpose of these comparisons is not to claim that all baselines have identical information access, but to separate three effects: model-free policy learning, direct simulator-level policy optimization, and global equation-driven solution methods.

**PPO.** PPO (Schulman et al., 2017) is included as a representative *model-free actor–critic baseline*. Its critic is trained through TD/Bellman-error-style updates, and the policy is updated through clipped policy optimization. This baseline tests whether the model-based information used by PG-DPO translates into a practical performance gain over a standard model-free RL method. In Cases 2–3, PPO is given an explicit time input and is trained using the same decision-time-anchored Monte Carlo objective used for evaluation. Nevertheless, PPO does not explicitly enforce the diagonal, time-consistent stationarity condition, which is the relevant first-order structure in the time-inconsistent regimes.

**DPO (Stage I).** DPO, or Direct Policy Optimization, denotes the Stage I warm-up policy $u_{\theta^\star}$ before the Pontryagin projection. It is the most direct Bellman-free comparator to PG-DPO because it optimizes the same simulator-level Monte Carlo objective but does not use the Stage II structural correction. This baseline also has the same level of simulator information as PG-DPO, making it the cleanest ablation for isolating the contribution of the Pontryagin projection. Since its gradients are obtained by differentiating through the simulated objective, DPO can be substantially more informative and lower-variance than score-function-based unbiased policy-gradient estimators.

Stage I DPO also serves as a strong substitute for a pure Monte Carlo policy-gradient baseline. It is Bellman-free and does not use critic learning or Pontryagin projection, but its gradients are obtained by differentiating through the simulated objective rather than by score-function estimators. Therefore, its gradient signals are typically more informative and lower variance than those of unbiased likelihood-ratio policy-gradient methods. In our additional experiments, Stage I DPO can be tuned to approach PG-DPO-level accuracy in the multiplicative benchmark, where maximizing the simulated objective is aligned with the relevant notion of optimality. However, in the non-multiplicative benchmark, Stage I DPO remains close to PPO-level accuracy. This supports the interpretation that strong direct policy optimization can work well when objective maximization and optimality are aligned, but is insufficient when the time-inconsistent equilibrium condition introduces a diagonal stationarity requirement that is not enforced by objective maximization alone.

**PINN and Deep BSDE/BSVIE.** PINN (Raissi et al., 2019) and Deep BSDE/BSVIE (Han et al., 2018) are included as *global equation-driven* baselines. These methods require access not only to the model dynamics, but also to the corresponding model-derived equations, such as PDE, BSDE, or BSVIE characterizations. They then fit a global surrogate object over the

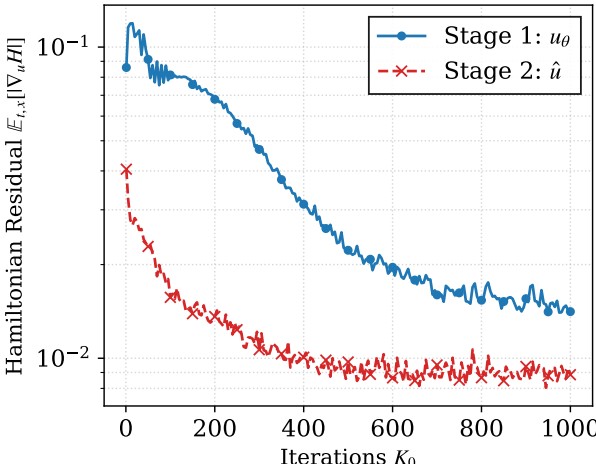

*Figure 6.* **Hamiltonian stationarity residual across iterations.** We plot the expected Hamiltonian residual $\mathcal{R} = \mathbb{E}[\|\nabla_u H\|_1]$ (log scale) during Stage 1 warm-up and after Stage 2 Adjoint-MC projection, while targeting Case 1 task 3.1.

state space and recover the control through first-order optimality conditions. Therefore, these baselines evaluate whether PG-DPO can remain competitive while bypassing global function fitting and avoiding accessing model-information of the full equation-level.

**Reference policies and omitted grid-based DP.** The label *Ground Truth* refers to numerical or analytic reference policies: the standard HJB reference in the multiplicative regime of Case 1, and the extended/equilibrium HJB reference in the time-inconsistent regimes of Cases 2–3. These references are used only for evaluation. We omit grid-based dynamic programming because its computational cost scales exponentially with the state dimension, making it impractical for the multidimensional settings considered in our experiments.

## E. Stage 2 Reduces the Hamiltonian Residual Despite Weak Stage-I Optimality

We report two curves in Figure 6: (i) *Stage 1 (warm-up)* Hamiltonian residual under the parametric policy $u_\theta$; and (ii) *Stage 2 (Adjoint-MC projection)* Hamiltonian residual under the projected control $\hat{u}$ obtained from the BPTT-estimated adjoint. A key observation is that even when Stage 1 is trained only coarsely (i.e., yields a weakly optimal and not yet PMP-consistent policy), the Stage 2 projection step still produces a pronounced reduction in the Hamiltonian residual. This indicates that the projection acts as a structure-enforcing correction: it can substantially tighten the PMP stationarity condition beyond what is achieved by additional warm-up alone, thereby empirically supporting that BPTT-based adjoint estimation combined with projection enforces the Bellman-free time-consistent optimality condition in practice.

## F. Scalability of PG-DPO

We further examine how the numerical accuracy of each method changes as the dimension increases. For each dimension $d \in \{10, 25, 50, 75, 100\}$, we generate a corresponding multi-asset market instance and evaluate the learned policies on the same $(\tau, W)$ grid used in the main experiments. The analytic log-utility solution is used as the reference policy. We report the grid-wise $L_1$ mean and standard deviation for both the portfolio policy $\pi$ and the consumption rate $c$. All results are shown on a logarithmic scale.

Figure 7 shows a clear separation between the projected PG-DPO policy and the other baselines. The Stage-I policy, denoted by DPO (Stage I), has relatively large errors across all dimensions, which suggests that amortized policy learning alone does not reliably recover the analytic structure in this high-dimensional control problem. By contrast, the PG-DPO projection reduces both $\pi$ and $c$ errors by several orders of magnitude and does not exhibit visible degradation over the tested range of dimensions.

The comparison also highlights a structural difference between the baselines. PINN and Deep BSVIE achieve moderate

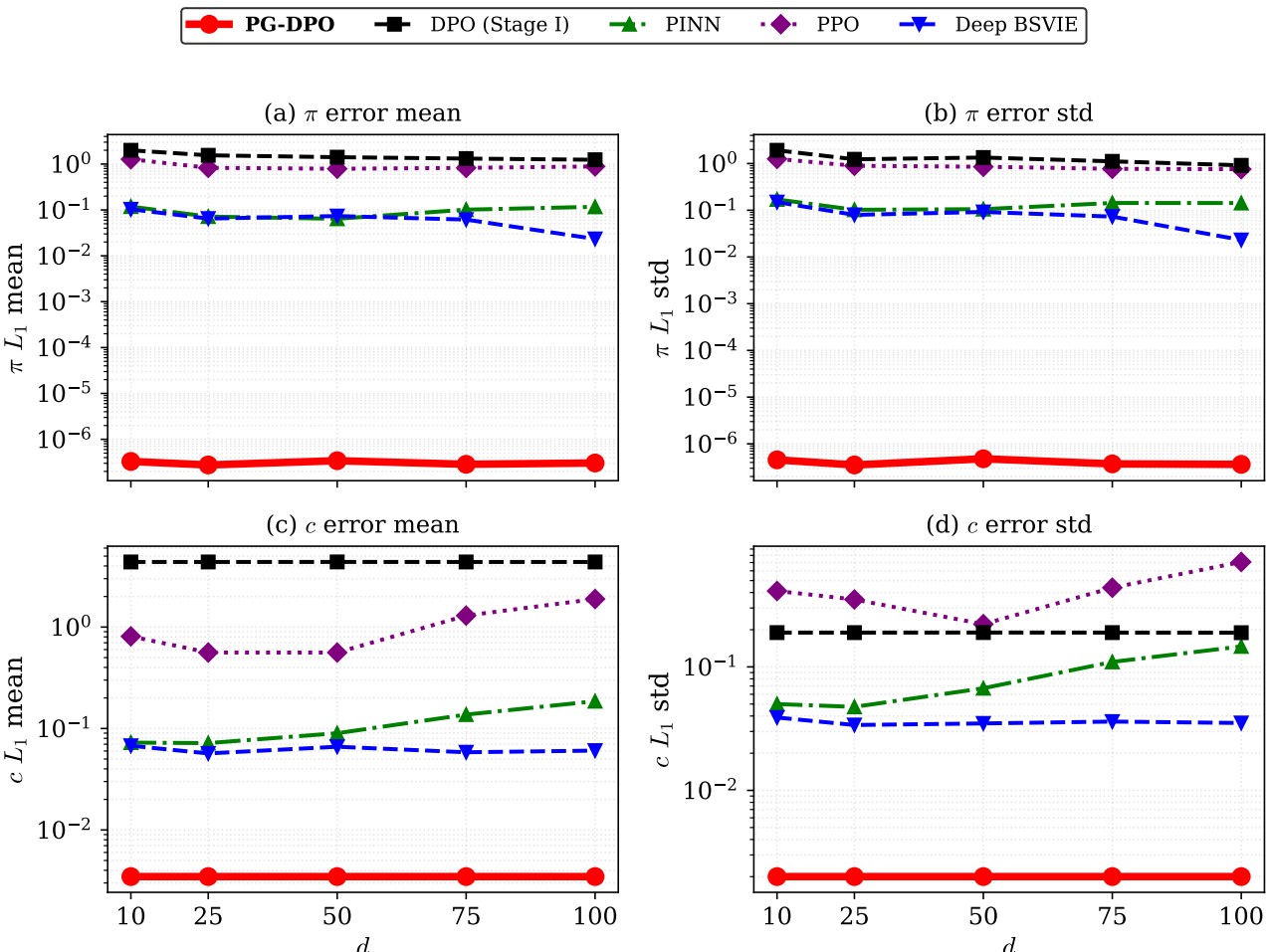

*Figure 7.* Dimension-sweep accuracy comparison. The horizontal axis denotes the portfolio dimension $d$, and the vertical axis reports the $L_1$ error against the analytic solution on a logarithmic scale. Panels (a) and (b) show the mean and standard deviation of the portfolio-policy error, while panels (c) and (d) show the corresponding consumption-rate errors. PG-DPO remains nearly flat as $d$ increases and stays several orders of magnitude below the purely amortized Stage-I policy and the learning-based baselines. This indicates that the Pontryagin projection substantially improves dimension-wise accuracy once the warm-up policy provides a usable local initialization.

errors for the portfolio component, but their consumption errors remain substantially larger than those of PG-DPO. PPO is less stable in the consumption component, especially in the standard deviation metric, which is consistent with the difficulty of learning continuous stochastic-control policies without directly enforcing the first-order optimality conditions. Overall, the dimension sweep supports the claim that the Pontryagin projection is not merely a low-dimensional correction: it provides a dimension-robust local refinement mechanism for this benchmark family.

## G. Computational Efficiency.

Table 4 summarizes the wall-clock runtime of the proposed PG-DPO inference in Stage 2. Our Method is computationally extremely efficient, requiring only **0.018 seconds** on a standard CPU and **0.03 seconds** on a GPU per query. This implies that our framework can support high-frequency decision-making (e.g., $> 50$Hz) on commodity hardware without requiring specialized accelerators.

As shown in Table 4, leveraging the massive parallelism of modern GPUs allows PG-DPO to maintain a **sub-second latency (0.17s)** even in these intensive settings. This confirms that the inference cost scales linearly with problem complexity ($\mathcal{O}(M \cdot N)$), making the method feasible for both lightweight deployment on CPUs and high-precision tasks on GPUs.

*Table 4.* Computational runtime analysis per query state $(t, s)$. We report the total wall-clock time for Stage 2 inference (including BPTT and action synthesis) across different hardware configurations, demonstrating sub-second latency even on CPUs.

| Hardware | Configuration | Horizon ($N'$) | Batch ($M_{MC}$) | Time / Query (s) |
|---|---|---|---|---|
| CPU (Intel Xeon) | Base | 16 | 256 | 0.018 |
| | Scalability Test A | 50 | 1024 | 0.25 |
| | Scalability Test B | 100 | 4096 | 1.80 |
| GPU (NVIDIA A100) | Base | 16 | 256 | 0.03 |
| | Scalability Test A | 50 | 1024 | 0.04 |
| | Scalability Test B | 100 | 4096 | 0.17 |

*Table 5.* **Case 1 budgets and hyperparameters.** We report (i) training update counts, (ii) batch sizes, (iii) rollout lengths, and (iv) major optimizer settings.

| Method | Budget / hyperparameters |
|---|---|
| PG-DPO (Stage 1 warm-up) | Adam steps: 500; batch trajectories: 256 per step; rollout steps $M = 64$; learning rate $10^{-3}$; grad clip 1.0. Variance reduction: antithetic ($\pm$) and Richardson extrapolation enabled; control variate enabled with EMA coefficient 0.98 and coefficient clipping 10.0. |
| PG-DPO (Stage 2 projection / costate) | For each time grid row ($N_T = 64$): costate repeats 256 with sub-batch 256. Each repeat uses antithetic pairing ($\pm$) when estimating pathwise sensitivities. |
| PPO | Iterations: 1000; parallel environments 256; rollout steps $M = 64$ (total env-steps: $1000 \times 256 \times 64 = 16.384M$). Update epochs: 5; minibatch size: 2048; learning rate $3 \cdot 10^{-4}$; GAE $\lambda = 0.95$; clipping $\epsilon = 0.2$; value coefficient 0.5; entropy coefficient 0.0. |
| PINN (KAN) | Adam steps 5000 with batch 4096; L-BFGS steps 1000 (strong Wolfe line search); learning rate $5 \cdot 10^{-4}$; terminal penalty weight 20; causal weighting parameter 100. |
| Deep BSDE | Adam steps 3000; batch 256; time steps 64; learning rate $10^{-4}$. |

# H. Additional Experimental Details and Baseline Fairness (Case 1)

## H.1. Compute budget and hyperparameters (Case 1)

Table 5 reports the exact training budgets and hyperparameters used in Case 1. These values are fixed *a priori* and applied consistently across random seeds.

**Network architectures.** We list the exact architectures used in Case 1 to prevent capacity mismatch claims:

- PG-DPO policy network: MLP with input dimension 2 and hidden width 128 for 2 layers (tanh activations), output dimension $m$.

- PPO policy: Gaussian MLP with shared trunk ($2 \rightarrow 164 \rightarrow 164 \rightarrow 164$), mean head $\mu(s) \in \mathbb{R}^m$ and global log-std parameter $\log \sigma \in \mathbb{R}^m$.

- PPO value network: MLP $2 \rightarrow 164 \rightarrow 164 \rightarrow 1$ (tanh).

- PINN (KAN): 3 Chebyshev KAN layers with hidden width 161 and degree 5.

- Deep BSDE: two MLPs (gradient network and $Y_0$ network) with 3 hidden layers of width 164 (tanh).

## H.2. PPO objective and implementation details

**Anchored reward shaping and $\gamma = 1$.** We define the per-step reward

$$r_k = \mathbf{1}[t_k < T] \cdot D(t_0, t_k) \cdot \ell(u_k) \cdot \Delta t, \tag{55}$$

and add the terminal reward at the final step:

$$r_{M-1} \leftarrow r_{M-1} + D(t_0, T) \, g(X_T). \tag{56}$$

Because discounting is already included via $D(t_0, \cdot)$ in $r_k$, PPO uses an undiscounted return with $\gamma = 1$. This removes ambiguity about how non-exponential discounting interacts with RL discount factors.

**GAE and targets.** Let $V_\phi(s_k)$ denote the value prediction. We compute generalized advantage estimates with $\gamma = 1$:

$$\delta_k = r_k + V_\phi(s_{k+1}) - V_\phi(s_k), \tag{57}$$

$$A_k = \delta_k + \lambda A_{k+1}, \qquad \lambda = 0.95, \tag{58}$$

and the value target is $\hat{R}_k = V_\phi(s_k) + A_k$. We normalize advantages within the collected batch.

**PPO surrogate.** The policy is Gaussian, $\pi_\theta(u|s) = \mathcal{N}(\mu_\theta(s), \mathrm{diag}(\sigma_\theta^2))$ with learnable $\log \sigma$. The clipped objective is

$$L^{\mathrm{clip}}(\theta) = \mathbb{E}\Big[ \min\big(\rho_k(\theta)A_k,\, \mathrm{clip}(\rho_k(\theta), 1 - \epsilon, 1 + \epsilon)A_k\big)\Big], \quad \rho_k(\theta) = \frac{\pi_\theta(u_k|s_k)}{\pi_{\theta_{\mathrm{old}}}(u_k|s_k)}, \tag{59}$$

with $\epsilon = 0.2$. The total loss is

$$\mathcal{L}(\theta, \phi) = -L^{\mathrm{clip}}(\theta) + c_v\,\mathbb{E}\big[(V_\phi(s_k) - \hat{R}_k)^2\big] - c_e\,\mathbb{E}\big[\mathcal{H}(\pi_\theta(\cdot|s_k))\big], \tag{60}$$

with $c_v = 0.5$ and $c_e = 0.0$.

**Evaluation policy.** At evaluation time, PPO uses the deterministic mean action $u = \mu_\theta(s)$ (no sampling), evaluated on the same $(t, x)$ grid as other methods.

