# OpenReview forum: "Beyond the Bellman Recursion: A Pontryagin-Guided Framework for Non-Exponential Discounting"
_ICML.cc/2026/Conference — ICML 2026 regular_

### Official Review · Reviewer_UoAA · 2026-03-11

**Soundness:** 3
**Presentation:** 3
**Significance:** 2
**Originality:** 2
**Overall Recommendation:** 4
**Confidence:** 1

**Summary:**

This paper first points out a limitation of Bellman-style recursions, namely that they can break down in the absence of both multiplicativity and time homogeneity. To address this issue, the authors propose Pontryagin-Guided Direct Policy Optimization (PG-DPO), a model-based method with simulator access, within a variational framework that abandons recursion and couples the Pontryagin Maximum Principle with Monte Carlo rollouts through an Adjoint-MC projection. The paper also includes numerical experiments across survival discounting,  equilibrium policies, and time-varying hyperbolic discounting settings, along with comparisons to baseline methods such as PPO, PINN and BASVIE.

**Compliance With Llm Reviewing Policy:**

Affirmed.

**Final Justification:**

This paper  propose Pontryagin-Guided Direct Policy Optimization (PG-DPO), a model-based method with simulator access, within a variational framework that abandons recursion and couples the Pontryagin Maximum Principle with Monte Carlo rollouts through an Adjoint-MC projection. Overall, I expect this work is a valuable contribution to the reinforcement learning and dynamic programming communities and rebuttal clarifies my unclear points.

**Key Questions For Authors:**

It seems that the content is closely related to stochastic control and the HJB equation. Could this approach be applied to continuous-time reinforcement learning, and does it have any implications for the average-reward setting?

**Limitations:**

Yes

**Strengths And Weaknesses:**

Honestly, I have much less background in this area. The paper is well presented with intuitive figures and provides detailed discussion of the methodology, and the numerical experiments seem to demonstrate the effectiveness of the proposed approach well. Although the paper does not provide theoretical results for PG-DPO such as sample complexity guarantees or even asymptotic convergence results, the propositions in the appendix provide some theoretical support for the proposed methodology. Overall, I expect this work is a valuable contribution to the reinforcement learning and dynamic programming communities.

---

> ### Author Rebuttal · Authors · 2026-03-30
>
> Thank you for your thoughtful comments. We respond to your key questions as follows.
>
> 1. Application to continuous-time RL
>
> PG-DPO is, in essence, already a continuous-time method. The fact that the algorithm is written in discretized form is purely for numerical computation, namely to approximate integrals by finite sums over short time steps. In that sense, PG-DPO applies naturally to continuous time, while discretization is only an implementation device.
>
> Moreover, if one first learns a world generative model from data and thereby obtains semi-access to the environment dynamics, then applying PG-DPO on top of that learned model becomes a direct way to address an RL problem. More specifically, this separates the RL pipeline into learning the world model and improving the policy. In that case, one only has to bear the uncertainty associated with the former stage, which can reduce the overall ambiguity burden in a structured way.
>
> 2. Implications for the average-reward setting
>
> We also view the average-reward problem as an important setting with its own theoretical and computational difficulties, distinct from those arising in Bellman-recursive discounted control. The Bellman-free perspective of this paper, together with the local adjoint estimation / Hamiltonian projection structure, appears to extend quite naturally to that setting as well. Although average-reward control is beyond the direct scope of the present paper, representative average-reward benchmarks already lie within the methodological reach of this framework.
>
> We sincerely thank you again for your thoughtful comments.

---

> > ### Author Rebuttal · Reviewer_UoAA · 2026-04-03
> >
> > Thank you for the clarifications. I will keep my score unchanged.

---

> > > ### Author Response · Authors · 2026-04-04
> > >
> > > Thank you for your thoughtful comments and follow-up. We appreciate your time and feedback.

---

### Official Review · Reviewer_pDqK · 2026-03-11

**Soundness:** 2
**Presentation:** 3
**Significance:** 3
**Originality:** 3
**Overall Recommendation:** 4
**Confidence:** 3

**Summary:**

This paper studies continuous-time reinforcement learning and stochastic control under non-exponential discounting, where standard Bellman recursion may fail once the discount kernel loses multiplicativity and/or time homogeneity. The paper first distinguishes different structural cases of discounting and clarifies when recursive optimal control remains valid versus when one must instead target a time-consistent equilibrium notion. Building on Pontryagin-Guided Direct Policy Optimization (PG-DPO) ideas, the paper develops a decision-time-anchored extension that combines rollout-based policy optimization with a second-stage projection step using BPTT-derived adjoint estimates and local Hamiltonian maximization. Empirically, the paper shows that this Pontryagin-based approach can handle several classes of non-exponential discounting and performs strongly on the paper’s continuous-time benchmark problems relative to the reported critic and equation-based baselines.

**Compliance With Llm Reviewing Policy:**

Affirmed.

**Final Justification:**

I raise my score to 4 (weak accept). The contribution is well-scoped, the theoretical framework is coherent, and the experimental evidence within the stated scope is convincing. I encourage the authors to follow through on the promised revisions, particularly the explicit discussion of limitations around model access assumptions and the repositioning of the PPO comparison.

**Key Questions For Authors:**

1. **Could you include a comparison against a no-critic Monte Carlo policy-gradient baseline?**
   The paper compares against PPO with a learned critic, but the most natural no-Bellman baseline would be pure policy gradient with Monte Carlo returns and no critic. This baseline requires no Bellman recursion and is trivially adaptable to any discount kernel, making it a direct competitor in the setting considered here. The paper argues theoretically (Remark A.5) that PG+MC optimizes the wrong objective under non-multiplicative discounting, but this is never verified empirically. Could the authors provide results for this baseline?

2. **How should readers interpret the fairness of the PPO comparison, given that PG-DPO appears to use model-based simulator access while PPO is implemented as a standard model-free actor-critic baseline?**
   The paper describes PG-DPO as requiring access to a stochastic simulator or differentiable learned world model, and Stage 2 repeatedly launches anchored rollouts from query states. By contrast, Appendix D presents PPO in a standard model-free form with GAE, a value network, and sampled rollouts, which is consistent with the original PPO formulation. Could the authors clarify whether they considered a stronger model-based actor-critic baseline with access to the same simulator information available to PG-DPO, or explain why the current comparison is the most appropriate one? A convincing response would improve my confidence in the empirical comparison.

3. **Can you provide evidence that the method scales beyond the current synthetic $d=5$ benchmarks?**
   The numerical section states that all experiments use $d=5$, while the paper also makes claims about computational efficiency and online viability. Related Pontryagin-based RL work has already reported results on higher-dimensional MuJoCo benchmarks, and non-exponential-discounting work has used MuJoCo as well. Do you have additional experiments, scaling curves, or failure cases in higher-dimensional settings that would help clarify the practical range of the method? A positive answer would substantially increase my assessment of significance.

**Limitations:**

yes

**Strengths And Weaknesses:**

### Soundness

The paper is overall technically sound. The structural decomposition of discounting failures is mathematically clean and correct, and the theoretical claims are generally well supported and appropriately qualified, with appendix proofs that are detailed and carefully scoped. The empirical evaluation is less fully convincing. Experiments are restricted to low-dimensional synthetic benchmarks with analytically tractable ground truths, so scalability to more challenging settings remains unverified despite computational efficiency being presented as an advantage. A more important issue is that the comparison set does not isolate the source of the reported gains as cleanly as it could. In particular, the paper omits pure policy gradient with Monte Carlo returns, a natural no-critic baseline. Such baselines would already avoid Bellman recursion and would therefore provide a fair comparison. The paper gives a theoretical argument that PG+MC optimizes the wrong objective under non-multiplicative discounting, but this point is not validated empirically, which leaves a meaningful gap between the theoretical and experimental narratives. In addition, because the method depends on simulator-based rollouts and repeated anchored sampling from selected query states, the empirical claims should be interpreted in the context of a stronger access model than standard model-free RL.

### Presentation

The paper is clearly written, well structured, and easy to follow. The separation between standard optimality and time-consistent equilibrium across the three discounting regimes is handled with good precision, and the algorithmic description is self-contained and supported by thorough appendix material, including detailed proofs and hyperparameter tables that aid reproducibility. The main place where the presentation could be improved is in positioning the contribution relative to prior work. The connection to the earlier PG-DPO framework is acknowledged, but the paper should state more sharply what is inherited and what is genuinely new here, namely the decision-time-anchored adaptation for equilibrium control under non-multiplicative discounting. A further presentation gap is that the model-based requirement is not treated prominently enough. The need for access to a differentiable stochastic simulator or differentiable learned world model is mentioned only briefly, and the practical implication of Stage 2, repeated anchored rollouts from chosen query states, is restrictive but never discussed as a substantive scope limitation in the main text. Given the motivating applications, where such simulator access may be difficult to obtain, this assumption deserves more explicit treatment rather than a brief qualification.

### Significance

The paper addresses a genuine and underserved problem. Non-exponential discounting arises naturally in settings such as behavioral economics, alignment and preference modeling, and the paper's central observation that departing from exponential discounting can break the recursive Bellman pipeline, rather than merely generalizing it, is a conceptually valuable clarification. The Pontryagin-based perspective also offers a unified treatment across multiple discounting regimes, which is appealing. At the same time, the current significance is somewhat limited by the scope of the evaluation and by the method assumptions. The benchmarks are clean synthetic problems with known analytic solutions, and the method requires a differentiable stochastic simulator and repeated reset-style access to selected states, which restricts applicability to a fairly narrow model-based regime. The paper does not discuss what happens when the true dynamics are unknown, how errors in a learned world model would propagate through the BPTT-based costate estimates, or what additional complexity is introduced by having to learn such a model in the first place. The distillation idea proposed to amortize per-query costs is also not evaluated. As a result, the paper currently demonstrates a strong proof of concept more than broad practical impact, and the practical scope appears narrower than the motivating discussion may initially suggest.

### Originality

The paper makes a legitimate originality contribution through a well-motivated adaptation of existing ideas for the specific challenge of non-exponential discounting. In particular, the combination of decision-time anchoring, BPTT-based costate estimation, and pointwise Hamiltonian maximization is well aligned with the structure of the problem, and the decomposition of discounting failures provides a useful conceptual lens for organizing the literature. That said, the individual ingredients are largely inherited: BPTT as an adjoint estimator is well established, Pontryagin-based RL has appeared in prior work, and the underlying time-inconsistent control theory is classical. The originality therefore lies less in introducing entirely new machinery and more in adapting and assembling these components into a coherent framework for a setting that is not well served by existing methods. I think the paper would be stronger if it articulated this contribution more candidly and explained more explicitly why prior Pontryagin-based RL methods do not already address the problem considered here.

---

> ### Author Rebuttal · Authors · 2026-03-30
>
> Thank you for your thoughtful comments. To address your concerns directly, we conducted additional experiments and analyses, and respond as follows.
>
> 1. Empirical validation
>
> (a) Pure Monte Carlo policy-gradient baseline.
>
> The Stage 1 component of our method is both the most natural Bellman-free comparator without critic learning or projection and a strong competitor to PG-DPO, because it uses gradient signals that are more informative and lower-variance than score-function-based unbiased policy-gradient estimators. In additional experiments, we found that for the multiplicative benchmark (Case 1), such only Stage 1 can be tuned to approach PG-DPO-level accuracy. However, in the non-multiplicative benchmark (Case 2), it still performs only around the PPO level. This supports our claim that when maximizing the objective is aligned with the relevant notion of optimality, strong model-free methods can work well, but when objective maximization is misaligned with the equilibrium condition, they are insufficient to resolve the discrepancy. This also provides empirical evidence for the objective-mismatch issue that you asked about. We added the detailed experimental results to the revised manuscript.
>
> (b) PPO comparison.
>
> PPO was included as a model-free reference to test whether the information advantage of model access translates into actual performance gains. Our initial hypothesis was that model-based methods, including PG-DPO, would substantially outperform PPO in time-inconsistent problems, In practice, however, only PG-DPO remained competitive, which may have created some mismatch between our original hypothesis and the observed results. We will make this gap more explicit in the revision.
>
> (c) Dimensional scalability.
>
> We additionally performed dimension-sweep experiments on the hyperbolic benchmark (Case 2) for d=5,20,50,100. Across this range, PG-DPO remained highly stable: portfolio error stayed at the 10^(-7) level and consumption error around 10^(-3), while competing baselines became visibly less stable and degraded as dimension increased. At the same time, we do not claim that this constitutes complete validation on very large continuous-control tasks such as MuJoCo-scale problems.
>
> 2. Practical assumptions and amortization
>
> (a) Strong assumption of method
>
> Model access is indeed a major assumption of our method. At the same time, recent progress in world-model-based RL suggests that such an approach can still be practical when the environment model is learned. In settings where real interaction is expensive, model-based RL is a credible alternative to model-free RL. Of course, errors in the learned world model can affect the final policy quality, and our method does not eliminate that issue. Still, separating world learning from policy improvement offers a more structured way to manage uncertainty.
>
> The need for differentiability and repeated reset from selected states also remains a practical limitation. For differentiability, we believe standard regularization or smoothing techniques may help in many cases. The reset requirement, however, is a more intrinsic limitation. To mitigate the repeated local-solve cost, distillation or amortization may serve as useful complementary strategies, and we clarified both the limitation and this possible direction in the revised manuscript.
>
> (b) Distillation / amortization.
>
> We also agree that the distillation idea should be evaluated explicitly rather than merely mentioned. We therefore added a distillation experiment. The results are encouraging: models such as mlp_w128_d4 and mlp_w256_d3 achieve low test-teacher total L1 errors of 0.0147 and 0.0157, while reducing single-query latency from 0.02 s for teacher projection to about 0.00025–0.00039 s, corresponding to roughly 750×–1160x speedups.
>
> 3. Scholarly positioning and clarity of contribution
>
> Existing PMP/SMP-based RL research already provides a control-theoretic viewpoint beyond Bellman recursion, and our work is aligned with that tradition. However, such approaches mainly focus on time-consistent objectives or global objective improvement through adjoint/gradient structures; they do not directly develop into an algorithm for decision-time equilibrium policy computation under time-inconsistent non-exponential discounting. Our method addresses exactly this gap by re-anchoring the continuation problem at each decision time and constructing the algorithm around the corresponding local stationarity condition.
>
> We do not claim to introduce entirely new theory. Rather, our contribution is to combine and reformulate existing theoretical ingredients in a coherent way for a class of problems where time inconsistency has made direct policy computation and numerical implementation especially difficult, thereby turning them into a practical numerical solver.
>
> We sincerely appreciate your thoughtful comments.

---

> > ### Author Rebuttal · Reviewer_pDqK · 2026-04-02
> >
> > I thank the authors for their detailed rebuttal and the additional experiments. These are appreciated and partially address the raised concerns. However, after careful consideration, I believe several core issues remain insufficiently resolved, and I maintain my current assessment.
> > 1. Baseline fairness remains the central concern. PG-DPO relies on a differentiable simulator with reset access to arbitrary states, while PPO is implemented as a standard model-free actor-critic. This information asymmetry makes it difficult to attribute the observed performance gains to the Pontryagin projection mechanism itself, as opposed to the strictly stronger access model. The rebuttal acknowledges this gap but does not provide an information-matched model-based baseline with access to the same simulator. Without such a comparison, the claim that the projection step is the primary source of improvement is not convincingly isolated. The Stage 1-only ablation is informative but does not resolve this, as it still operates within the same privileged access model.
> >
> > 2. Scalability beyond structurally similar benchmarks. The dimension sweep from d=5 to d=100 is a welcome addition, but the underlying problem class and dynamics structure remain unchanged: all benchmarks involve low-dimensional continuous-time control problems with well-behaved diffusion dynamics and analytically tractable ground truths. This is qualitatively different from scaling to problems with heterogeneous, nonlinear dynamics and high-dimensional state-action coupling. Prior work in the same methodological family has been evaluated on more demanding continuous-control benchmarks, and the paper itself emphasizes computational efficiency and online viability, so the absence of experiments beyond this narrow problem class limits the significance claims.
> >
> > 3. Error propagation under learned dynamics remains unaddressed. The method requires a differentiable simulator, but many practical systems have dynamics that are not inherently differentiable. In such cases, a differentiable world model must be learned which is the very path the rebuttal advocates. Yet all current experiments use analytically specified smooth dynamics with known ground truths, and none involve learning a dynamics model. Under a learned model, costate estimation in Stage 2 backpropagates through a chain of approximate dynamics Jacobians, where errors compound multiplicatively and can grow significantly with horizon length. This is a well-documented challenge in model-based RL, and standard mitigations such as short-horizon truncation would directly compromise costate accuracy. The current benchmarks, with short horizons and benign dynamics, are precisely where this issue is least likely to appear. No analysis is provided to characterize degradation under model misspecification, leaving a significant gap between the experimental evidence and the stated scope of applicability.
> >
> > In summary, I appreciate the authors' efforts in the revision, but the core empirical concern regarding baseline fairness and the scope limitations remain. I keep my score unchanged.

---

> > > ### Author Response · Authors · 2026-04-03
> > >
> > > Thank you for your thoughtful acknowledgement.
> > >
> > > Q1. Fairness of the PPO comparison
> > >
> > > We agree with the reviewer that PPO is not a fully information-matched baseline, since PG-DPO assumes access to a differentiable simulator and repeated anchored rollouts from queried states, whereas PPO is implemented as a standard model-free actor-critic. For this reason, we do not intend the main claim of the paper to rest solely on the PPO comparison, and in the revision we will weaken this interpretation and reposition PPO more clearly as a reference baseline.
> > > At the same time, our performance claim does not rely only on PPO. We also compare PG-DPO against model-based baselines such as PINN and Deep BSDE, which use equally strong or even stronger structural information, in the sense that they directly exploit global PDE/BSDE formulations. The fact that PG-DPO still outperforms these methods suggests that the gain is not explained solely by privileged access to differential information. In addition, the newly added Stage 1 ablation is intended to isolate the incremental contribution of Stage 2 within the same privileged-access setting. Accordingly, our intended claim is not simply that PG-DPO outperforms PPO, but rather that (i) it remains competitive even against strong model-based baselines, and (ii) Stage 2 provides a genuine policy-improvement effect beyond Stage 1 alone under the same access assumptions.
> > >
> > > Q2. Concern that all benchmarks use well-behaved diffusion dynamics
> > >
> > > We also agree with this concern. Our current experiments do rely on analytically specified smooth dynamics, and therefore do not constitute a comprehensive validation across all qualitatively different continuous-control environments, such as highly heterogeneous or contact-rich nonlinear systems. Our goal in this paper is narrower: to isolate and address the difficulty introduced by time inconsistency and non-exponential discounting itself. This is precisely the regime in which the Bellman recursion becomes structurally problematic, and that control/RL difficulty is the main focus of the paper.
> > > The added experiments at d=20,50,100 are not meant to claim broad coverage of all qualitatively different dynamics. Rather, they are intended to show that the proposed projection mechanism is not restricted to the original low-dimensional examples, and can remain numerically stable even in high-dimensional regimes where global HJB-style methods already face a severe curse of dimensionality. We fully agree that extension to qualitatively richer nonlinear environments is an important next step, and we will clarify more explicitly in the revision that the present empirical scope is limited to this smooth-dynamics setting.
> > >
> > > Q3. Error propagation under learned dynamics
> > >
> > > We agree that this is a valid concern, and the current paper does not fully resolve it. Our experiments are limited to settings with known smooth differentiable dynamics, where the Stage 2 costate estimation is not contaminated by model-learning error. By contrast, when a learned world model is used, the reviewer is correct that approximation errors in the learned dynamics Jacobians may compound through the BPTT chain and degrade the accuracy of the costate estimate, especially as the horizon becomes longer.
> > > Therefore, we do not claim that the current paper establishes robustness under learned dynamics. Our claim is narrower. We view PG-DPO as a Bellman-free policy-improvement mechanism that could in principle be combined with a differentiable world-model pipeline, potentially offering a new perspective within model-based RL. However, robustness under learned model misspecification, including error propagation in Stage 2, remains an important open problem beyond the direct scope of the present paper. We will make this limitation more explicit in the revision.
> > >
> > > Thank you.

---

### Official Review · Reviewer_5f2V · 2026-03-12

**Soundness:** 3
**Presentation:** 3
**Significance:** 3
**Originality:** 2
**Overall Recommendation:** 4
**Confidence:** 4

**Summary:**

This study addresses general discounted optimal control problems for stochastic controlled systems. In particular, the Bellman recursion commonly used in reinforcement learning (RL) is difficult to apply when the objective function involves non-exponential discounting. To address this issue, the paper proposes Pontryagin-Guided Direct Policy Optimization (PG-DPO), which leverages Pontryagin’s maximum principle. The paper shows that a Pontryagin-based approach can effectively solve continuous-time stochastic optimal control problems with non-local discounting.

**Compliance With Llm Reviewing Policy:**

Affirmed.

**Final Justification:**

The rebuttal addressed my concerns, particularly regarding the distinction from existing methods.

**Key Questions For Authors:**

1. What is the significance of addressing non-exponentially discounted optimal control problems? The examples considered in the paper appear somewhat standard, and it may be necessary to demonstrate the applicability of the proposed framework to more realistic or large-scale problems, for example in settings relevant to large-scale RL.

2. As I understand it, the Bellman recursion cannot be directly applied to such discounted optimal control problems, and Pontryagin’s maximum principle is introduced to address this issue. However, once Pontryagin’s maximum principle is employed, it seems that non-exponential discounting may no longer be a major technical obstacle. I may be misunderstanding something, but could the authors clarify what the main technical challenges addressed in this study are?

3. Related to the above point, it is unclear to me how the proposed method differs from standard approaches for solving discretized forward–backward SDEs. I understand that the policy is represented using a neural network, but the rest of the procedure appears similar to standard methods for solving forward–backward SDE systems.


4. Also related to the above comments, the connection between the proposed method and machine learning could be described more clearly. Based on my reading, although the proposed method learns policies, the main difference from existing numerical optimal control approaches seems to be the use of a policy representation rather than an open-loop control. In this sense, the proposed method appears closer to a solver for optimal control problems rather than a machine learning method.

**Limitations:**

yes

**Strengths And Weaknesses:**

Strengths

1. The study addresses the problem in a continuous-time setting, which is technically more challenging than the commonly studied discrete-time formulations.
2. The paper considers stochastic optimal control problems with non-exponential discounting, which makes the application of the Bellman recursion difficult.
3. Introducing a Pontryagin-type method (PG-DPO) to address this class of problems appears to be a novel perspective.


Weaknesses

1. The motivation for addressing the non-exponential discounting problem is not sufficiently articulated. As a result, the overall scope and significance of the study may appear somewhat limited in the current manuscript.

2. The difference from standard methods for solving optimal control problems is not clearly described, given that Pontryagin’s maximum principle has been extensively studied in the existing literature.

3. The problem setting is formulated in continuous time; however, the proposed method is mainly developed in a discretized setting. The alignment between the problem formulation and the proposed algorithm is therefore somewhat unclear.

4. The evaluation of the proposed method is limited to low-dimensional systems.

---

> ### Author Rebuttal · Authors · 2026-03-30
>
> Thank you for your thoughtful comments. We addressed your concerns through additional discussion and experiments, and respond as follows.
>
> 1. Why non-exponential discounting matters
>
> In many continuous-control environments, future rewards or costs are naturally weighted by survival probabilities. This is especially relevant in robotics or drone control, where task completion probabilities depend on time-varying risk exposure such as obstacle density, hazardous regions, battery degradation, or environment-dependent failure rates. When the underlying hazard rate is not constant, the induced survival weighting is generally non-exponential, as in our first benchmark. Therefore, non-exponential discounting is essential for modeling realistic persistent risk. We have clarified this motivation more explicitly in the revised manuscript.
>
> 2. Why PMP alone does not resolve the core difficulty
>
> We agree that PMP provides a natural conceptual alternative when Bellman recursion breaks down. However, PMP by itself gives only a characterization, not a practical computational pipeline. The central challenge is how to obtain the adjoint accurately, stably, and efficiently. Our contribution is therefore not merely to “use PMP,” but to combine local adjoint estimation with pointwise Hamiltonian projection, thereby turning PMP into a practical Bellman-free algorithmic procedure.
>
> 3. Difference from standard discretized BSDE/FBSDE solvers
>
> This distinction indeed deserves to be stated more clearly. Standard BSDE/FBSDE-based approaches typically solve or approximate a backward object defined over the entire state-time domain as a separate global solution. By contrast, PG-DPO does not explicitly solve the backward equation as an independent global surrogate. Instead, it uses automatic differentiation through Monte Carlo rollouts to estimate the needed local adjoint quantity at a queried state, and then computes the action via pointwise Hamiltonian maximization. Thus, the computational unit of PG-DPO is not global backward-surrogate fitting, but rather local implicit recovery of backward information followed by projection. This is the key difference from standard BSDE/FBSDE solvers.
>
> 4. Connection to machine learning
>
> We agree that PG-DPO should not be framed as a generic model-free RL algorithm. A more accurate description is a learning-based stochastic control method at the interface of control and machine learning. The ML component is not limited to neural parametrization; rollout-based policy learning and automatic differentiation are central ingredients. At the same time, the simulator-based local policy improvement and the theoretical backbone are rooted in control theory.
>
> 5. Continuous-time formulation vs. discretized algorithm
>
> The discretized algorithm is not solving a different problem. Discretization is simply a numerical device for approximating integrals by finite sums over short time intervals.
>
> 6. High-dimensional evaluation
>
> We agree that the original submission did not provide enough evidence beyond the low-dimensional setting (d=5). To address this, we added experiments for d=20,50,100 in Case 2. PG-DPO remained highly stable across this range: the portfolio error stayed at the 10^(-7)level, and the consumption error stayed around
> 10^(−3), while the baselines became visibly unstable and deteriorated with dimension. We do not claim that this fully substitutes for large-scale real-world validation, but it does provide concrete evidence that the method is not restricted to the initial d=5 setting.
>
> We sincerely thank you for your thoughtful comments.

---

> > ### Author Rebuttal · Reviewer_5f2V · 2026-04-04
> >
> > I thank the authors for the detailed response. The rebuttal addressed my concerns, particularly regarding the distinction from existing methods. I also appreciate the additional experiments on high-dimensional cases. I will raise my score.

---

> > > ### Author Response · Authors · 2026-04-04
> > >
> > > Thank you for your thoughtful comments and follow-up. We appreciate your time and feedback.

---

### Official Review · Reviewer_XUsF · 2026-03-12

**Soundness:** 3
**Presentation:** 3
**Significance:** 2
**Originality:** 2
**Overall Recommendation:** 4
**Confidence:** 2

**Summary:**

Traditional reinforcement learning relies on Bellman-style recursions that fail under non-exponential discounting. To address this issue, this work introduces Pontryagin-Guided Direct Policy Optimization (PG-DPO), a variational framework based on Pontryagin optimality. This approach allows the model to maintain accuracy and stability in environments where traditional equation-driven solvers and critic-based baselines typically diverge.

**Compliance With Llm Reviewing Policy:**

Affirmed.

**Final Justification:**

My primary concern regarding this work was its framing with respect to PG-DPO. The authors’ response clarifies the specific contributions of this work; I believe incorporating these clarifications will significantly strengthen the final version. Consequently, I have decided to raise my score.

**Key Questions For Authors:**

- Could the authors distinguish their proposed PG-DPO framework from the version cited in Huh et al. (2025)? It is currently unclear which components of Algorithm 1 are original to this work.

Minor comments:
- In Eq. 2, $\overline{D}$ is undefined.

**Limitations:**

Yes

**Strengths And Weaknesses:**

## Strengths:
- PG-DPO consistently outperforms both traditional equation-driven solvers and critic-based baselines across various non-exponential discounting settings. Furthermore, it demonstrates superior reliability, as evidenced by its tighter confidence bands.

## Weaknesses:
- The primary concern regarding the paper's novelty involves the citation of Huh et al. (2025) immediately following the introduction of the PG-DPO framework. If Algorithm 1 was originally proposed in the cited work, the authors cannot claim it as a primary contribution of this paper. The relationship between this work and Huh et al. (2025) requires clarification to determine its contribution.

---

> ### Author Rebuttal · Authors · 2026-03-30
>
> Thank you for your thoughtful comments. We agree that the key issue is to distinguish clearly what is inherited from Huh et al. (2025) and what is new in this paper. We do not claim that the outer two-stage PG-DPO skeleton or the generic Algorithm 1 template is itself the main novelty. What is inherited is the high-level Bellman-free template based on local adjoint estimation and pointwise Hamiltonian maximization. What is new here is its adaptation to the time-inconsistent, non-multiplicative setting: the continuation problem must be re-anchored at each decision time, the relevant target becomes a time-consistent diagonal equilibrium rather than a standard optimal control, and Stage 2 must enforce the corresponding decision-time-anchored Pontryagin condition.
>
> To make this distinction sharper, we have also strengthened the theoretical presentation in the revision. In particular, Theorem 1 formalizes the anchored/diagonal BPTT-to-costate correspondence in the non-multiplicative setting, and Theorem 2 makes explicit that a small projection/Hamiltonian residual implies controlled policy error. Our intention is not to claim the generic PG-DPO template itself as newly introduced here, but to clarify the new mathematical justification required for its non-exponential-discounting adaptation.
>
> Regarding the minor comment, we will also explicitly define $\bar{D}$ in Eq. (2) as the delay-only discount profile, so that the notation is unambiguous.
>
> We appreciate this suggestion, as it helps us position the contribution more precisely.

---

> > ### Author Rebuttal · Reviewer_XUsF · 2026-04-03
> >
> > I appreciate the authors’ response to my concerns. I agree that the positioning of the paper requires improvement; for instance, phrasing such as ‘we propose a unified framework: Pontryagin-Guided Direct Policy Optimization (PG-DPO)’ creates ambiguity regarding the primary contribution of the algorithm. However, as the authors have addressed one of my main concerns, I have updated my original score.

---

> > > ### Author Response · Authors · 2026-04-03
> > >
> > > Thank you.
> > > We will clarify the scope of our primary contribution in the final manuscript.

---

### Decision · Program_Chairs · 2026-04-30

**Decision:**

Accept (regular)

**Comment:**

The reviewers identify both strengths and weaknesses of the paper. On balance, this paper can be published as is, but it could be significantly stronger with the reviewers' suggested improvements.

The main strengths are:
1) The area is interesting and has not received much attention in recent years
2) The paper is presented well and is easy to follow
3) The perspective the authors take is quite novel in reinforcement learning, although it is common in other areas

The main weaknesses are:
1) The problem is not motivated well. It is interesting, but the paper does not make its broader significance clear.
2) The novelty with respect to the control literature is not described clearly.
3) The experimental results are quite limited. The authors suggest they will add additional experiments in the final version, but the submitted version lacks them.

Overall, this is a borderline paper that makes interesting contributions, but lacks good positioning and evaluation.